# Simulating ice segregation and thaw consolidation in permafrost environments with the CryoGrid community model

Juditha Aga[1], Julia Boike[2,3], Moritz Langer[2,4], Thomas Ingeman-Nielsen[5], and Sebastian Westermann[1,6]

[1]Department of Geosciences, University of Oslo, Oslo, Norway
[2]Alfred Wegener Institute (AWI), Helmholtz Centre for Polar and Marine Research, Potsdam, Germany
[3]Geography Department, Humboldt-Universität zu Berlin, Berlin, Germany
[4]Department of Earth Sciences, Vrije Universiteit Amsterdam, Amsterdam, the Netherlands
[5]Department of Environmental and Resource Engineering, Technical University of Denmark, Lyngby, Denmark
[6]Centre for Biogeochemistry in the Anthropocene, University of Oslo, Oslo, Norway

**Correspondence:** Juditha Aga (juditha.aga@geo.uio.no)

**Abstract.** The ground ice content in cold environments influences the permafrost thermal regime and the thaw trajectories in a warming climate, especially for soils containing excess ice. Despite their importance, the amount and distribution of ground ice are often unknown due to lacking field observations. Hence, modelling the thawing of ice-rich permafrost soils and associated thermokarst is challenging as ground ice content has to be prescribed in the model set-up. In this study, we present a model

scheme, capable of simulating segregated ice formation during a model spin-up together with associated ground heave. It provides the option to add a constant sedimentation rate throughout the simulation. Besides ice segregation, it can represent thaw consolidation processes and ground subsidence under a warming climate. The computation is based on soil mechanical processes, soil hydrology by Richards equation and soil freezing characteristics. The code is implemented in the CryoGrid community model (version 1.0), a modular land surface model for simulations of the ground thermal regime.

The simulation of ice segregation and thaw consolidation with the new model scheme allows us to analyze the evolution of ground ice content in both space and time. To do so, we use climate data from two contrasting permafrost sites to run the simulations. Several influencing factors are identified, which control the formation and thaw of segregated ice. (i) Model results show that high temperature gradients in the soil as well as moist conditions support the formation of segregated ice. (ii) We find that ice segregation increases in fine-grained soils and that especially organic-rich sediments enhance the process. (iii)

Applying external loads suppresses ice segregation and speeds up thaw consolidation. (iv) Sedimentation leads to a rise of the ground surface and the formation of an ice-enriched layer whose thickness increases with sedimentation time.

We conclude that the new model scheme is a step forward to improve the description of ground ice distributions in permafrost models and can contribute towards the understanding of ice segregation and thaw consolidation in permafrost environments under changing climatic conditions.

# 1 Introduction

Permafrost underlies an area of approximately 14 million $\text{km}^2$, which corresponds to around 15 % of the exposed Northern Hemisphere or 11 % of the global exposed surface (Obu et al., 2019; Obu, 2021). Frozen soils with a high organic content are widespread (Hugelius et al., 2014) and these permafrost environments largely evolved in the last glacial cycle, accumulating massive amounts of soil organic carbon. Especially in Siberia and Alaska, they are often associated with a high content in excess ground ice (Saito et al., 2020; Schirrmeister et al., 2011). The climatic, sedimentological, geomorphological and ecological conditions in the past govern today's distribution of ground ice (Gilbert et al., 2016). This is especially true for syngenetic permafrost, which builds during sedimentation or formation of organic material in cold environments and is typically ice-rich. In contrast, epigenetic permafrost forms with a time lag to the sedimentation processes in the ground, and is often characterized by ice-poor soils (Gilbert et al., 2016). Ground ice can be present as pore ice or excess ice, which can occur as relict ice, wedge ice, segregated ice or injection ice (French and Shur, 2010; French, 2017).

Segregated ice is ground ice, which forms through migration of soil water to the frozen fringe (Harris et al., 1988). It occurs as discrete ice lenses or layers, which can range from less than a millimeter to more than 10 m (French and Shur, 2010; Harris et al., 1988). Segregated ice is typically associated with ground heave as the ice content in the ground increases (Fig. 1; Miller, 1972; Taber, 1929). It can build up in epigenetic permafrost when unfrozen soil water from the active layer is attracted towards the freezing front (Guodong, 1983; Mackay, 1983; Taber, 1929), accumulating ice near the permafrost table. Besides, ice segregation can occur at the base of the permafrost, as water is drawn from the soil below as permafrost is forming. However, the evolution of syngenetic permafrost can result in enhanced ice formation. Accumulation of organic material as well as sedimentation in alluvial, eolian or hillslope settings can lead to a rise in the permafrost table and hence a growth of segregated ice (Guodong, 1983; French and Shur, 2010). In this context, segregated ice can form also together with syngenetic ice wedge growth, forming ice lenses within polygonal permafrost as observed in Yedoma deposits (Schirrmeister et al., 2013).

Layers of segregated ice in the ground are widespread in permafrost environments, especially in fine-grained sediments, which are susceptible to ice segregation (French and Shur, 2010). Cryostratigraphic mapping has been performed in numerous studies, documenting segregated ice especially in Siberia (Andreev et al., 2009; Meyer et al., 2002; Schirrmeister et al., 2008; Siegert and Babiy, 2002) and North America (French et al., 1986; Heginbottom, 1995; Kanevskiy et al., 2013; Shur and Jorgenson, 1998). O'Neill et al. (2019) modelled the occurrence of segregated ice in Canada, showing abundance in fine-grained lacustrine sediments, raised peat plateaus and uplifted marine sediments. This distribution is supported by borehole information and field studies (Gaanderse et al., 2018; Smith et al., 2007; Wolfe and Morse, 2017). Observations of the cryostructure, including segregated ice, can reveal information about the evolution of the ground, e.g. if the permafrost was formed syngenetically or epigenetically. Furthermore, thaw unconformities, such as former active layers, can be detected by changes in the ice content with depth and the isotopic signature of the ground ice (French and Shur, 2010).

The ground ice content and its distribution strongly determines the sensitivity of permafrost to thaw (Jorgenson et al., 2010; Nitzbon et al., 2019). Ground ice formation releases latent heat, delaying the freezing. In contrast, ice-rich layers in the soil can delay permafrost degradation as energy is consumed upon melting of the ground ice, which is consequently not available

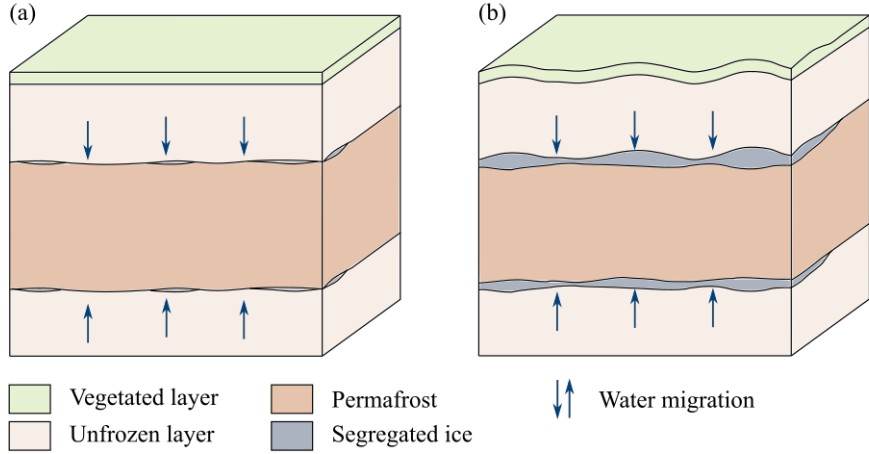

**Figure 1.** Illustration of ground heave through ice segregation at the top and the base of the permafrost. (a) Lenses of segregated ice are formed. (b) If the segregated ice is preserved over a long time period, layers with segregated (excess) ice are forming, causing heave of the ground surface. Figure modified after Fu et al. (2022).

for the warming of the ground (Riseborough, 1990). In addition, ice segregation can continue even under thawing conditions, forming ice layers at the top of the permafrost, continuously delaying the warming process. However, if excess ice is melted, it can result in thermokarst and ground subsidence (Farquharson et al., 2019; Kokelj and Jorgenson, 2013; Nitzbon et al., 2019). In consequence, substantial geomorphological changes reshape the landscape, manifested in the formation of lakes, thaw slumps, gullies and the transformation of low-centered to high-centered polygons (Kokelj and Jorgenson, 2013; Liljedahl et al., 2016; Nitzbon et al., 2019). These processes could contribute to accelerating the mobilization of permafrost carbon, which may further increase atmospheric carbon concentrations, a process known as the permafrost carbon feedback (Miner et al., 2022; Schuur et al., 2008). Furthermore, ground ice controls the hydrological and mechanical properties of the soil by reducing permeability and increasing the mechanical strength (Painter and Karra, 2014). These parameters control the structural stability of the ground. Upon thawing, mass movements along slopes might increase and together with ground subsidence, the reduced stability can endanger human infrastructure and settlements (Hjort et al., 2022; Schneider von Deimling et al., 2021).

As about 20 % of the permafrost in the Northern Hemisphere is prone to thermokarst processes (Olefeldt et al., 2016), it is highly important to develop models representing the formation and melt of (excess) ground ice, such as segregated ice, to better understand Arctic landscape evolution. This will improve our capabilities to assess soil stability under transient climate conditions, which is essential for the sustainable design of infrastructure in permafrost environments (Dumais and Konrad, 2018; Painter and Karra, 2014). Modelling of ground consolidation requires implementing soil mechanical processes in addition to heat and water transfer (Dumais and Konrad, 2018). Thaw consolidation has been the focus of model development for many decades (Morgenstern and Nixon, 1971; Nixon and Morgenstern, 1973; Sykes et al., 1974; Konrad and Morgenstern, 1980, 1981, 1982a, b; Konrad, 1983; O'Neill, 1983; Nixon, 1991; Foriero and Ladanyi, 1995; Dumais and Konrad, 2018), and

different approaches have been presented for modelling ice segregation (Fu et al., 2022; Fisher et al., 2020; Lacelle et al., 2022). An example is the approach of An and Allard (1995), who successfully demonstrated palsa formation through accumulation of segregated ice. However, these processes are typically not implemented in land surface models and simulating the long-term evolution of segregated ice has not been performed yet (Fu et al., 2022).

As a consequence, previous models targeting ground ice thaw and thermokarst require excess ice distributions prescribed from field observations, which makes applications at sites without ground ice data challenging. Furthermore, as ice segregation is not implemented, they neglect the delay in permafrost warming through the thaw of segregated ice layers, formed during the simulation period. In this study, we demonstrate that segregated ice in the ground can be simulated with a climate-dependent spin-up procedure, which aims at reproducing the evolution of ground ice stocks. We present a new model approach, which is capable of coherently simulating both ice segregation and thaw consolidation, based on heat and water transfer as well as soil mechanical processes. The model code is implemented within the framework of the modular CryoGrid community model (version 1.0, referred to as "CryoGrid" from here on), a land surface model for permafrost applications (Westermann et al., 2023). The main objectives of this study are the following:

- We demonstrate a proof of concept for simulating ice segregation and thaw consolidation with associated ground heave and subsidence. To do so, we run various model scenarios with climate data from two contrasting permafrost sites, representing cold continental and relatively warm maritime permafrost conditions.

- We evaluate the performance of our model to reproduce known controlling factors on ice segregation and thaw consolidation. Particularly, we analyze different climatic conditions (by applying different forcing data sets), the soil type (by using different grain sizes and compositions) and external loads.

- We investigate the capability of the model to simulate long-term ice segregation under a constant sedimentation regime.

## 2 Methods

In this work, we extend the capabilities of the CryoGrid community model (Westermann et al., 2023) with a representation of soil mechanical processes. The CryoGrid community model is a modular framework for simulating the permafrost thermal state and the water and ice balance. To set up simulations, the user can choose between different so-called "stratigraphy classes", which are characterized by specific model physics and state variables. As an example, one stratigraphy class can calculate soil water contents with a simple bucket model, while another can account for water redistribution through Richards equation in unsaturated soils. Furthermore, there are dedicated stratigraphy classes for non-ground materials, in particular for the seasonal snow cover. The stratigraphy classes can be stacked vertically, so that the available classes representing snow can can be flexibly combined with a range of classes for ground materials.

In this study, we introduce a new stratigraphy class denoted *GROUND_freezeC_RichardsEq_seb_pressure*, a fully fledged process model for soils. It is based on the already existing stratigraphy class *GROUND_freezeC_RichardsEq_seb* (Westermann et al., 2023) and inherits many of its functionalities. While a detailed description of the CryoGrid community model is provided

in Westermann et al. (2023), we summarize the main aspects relevant for this work in Sect. 2.1, before describing the defining equations and main properties of the new stratigraphy class. The model is demonstrated and evaluated for two field sites (Sect. 2.2), for which we simulate a range of model scenarios (Sect. 2.4) with different settings for subsurface properties and model parameters (Sect. 2.3).

## 2.1  Model description

In the stratigraphy classs *GROUND_freezeC_RichardsEq_seb*, each model grid cell is characterized by its volumetric contents of the mineral, organic, water and ice components, which also define the porosity. The upper boundary is defined as the interface between the ground surface and the atmosphere at which the surface energy balance is applied, controlled by the exchange of short-wave and long-wave radiation, as well as latent and sensible heat fluxes. To calculate the surface energy balance, the forcing data must prescribe time series of air temperature, solid and liquid precipitation, wind speed, short-wave and long-wave radiation, specific humidity and air pressure. The lower boundary condition (set at a user-specified depth) is defined by a constant geothermal heat flux.

Subsurface heat transfer is based on both conductive and advective fluxes. The calculation of heat conduction follows Fourier's law with the thermal conductivity of the material being the controlling factor. The heat transfer through advection is determined by vertical water fluxes.

A soil freezing characteristic describing the relationship between ground temperature and unfrozen water content (Painter and Karra, 2014) is implemented in *GROUND_freezeC_RichardsEq_seb*. To determine liquid water and ice contents in frozen soils, we first calculate the matric potential for unfrozen conditions $\psi_0$ [m] from which the matric potential in frozen state $\psi$ [m] and finally the water content can be inferred (assuming no residual water). A detailed description of the approach can be found in Westermann et al. (2023).

The water balance in *GROUND_freezeC_RichardsEq_seb* is based on vertical water flux $j_w^v$ [m s$^{-1}$] controlled by the Richards equation (Richards, 1931):

$$j_w^v = -K_w \left( \frac{\partial \psi}{\partial z} + 1 \right) \tag{1}$$

with $\psi$ [m] being the matric potential, $z$ [m] the vertical coordinate and $K_w$ [m s$^{-1}$] the hydraulic conductivity. The subscript $w$ denotes *water* and the superscript $v$ signifies *vertical* for model variables. The hydraulic conductivity is computed according to Van Genuchten (1980) as well as Hansson et al. (2004) to take into account ground ice blocking water-filled pores. To do so, a depth stratigraphy of the permeability $k_w$ [m$^2$] is defined by the user (see Westermann et al., 2023). Vertical water fluxes between unsaturated grid cells are initiated through gradients in matric potentials as well as gravitational potentials. In saturated unfrozen regimes, water flow by gravity occurs according to Darcy's law.

In addition to vertical water fluxes, the CryoGrid community model provides the option for lateral water transport, e.g. through overland flow (Westermann et al., 2023), which is induced according to the Gauckler-Manning equation (Gauckler, 1867; Manning et al., 1890). Depending on the Gauckler-Manning coefficient and the local ground surface gradient, standing

surface water is removed from the system. Other lateral water fluxes such as subsurface drainage are not included in the model setup.

The seasonal snow cover is implemented by applying the snow class *SNOW_crocus2_bucketW_seb*, which includes snow microphysics, the surface energy balance and snow hydrology. It is based on the Crocus snow scheme (Vionnet et al., 2012) and described in Westermann et al. (2023) and Zweigel et al. (2021). Excess water from snow melt is transferred to the uppermost grid cell of the new *GROUND_freezeC_RichardsEq_seb_pressure* class, which can represent standing surface water by a dedicated pool from which it can either run off laterally through Gauckler-Manning flow or infiltrate into the ground according to soil properties and moisture conditions.

While the general CryoGrid framework and all other stratigraphy classes (such as the snow class, see above) remain unchanged, the new stratigraphy class *GROUND_freezeC_RichardsEq_seb_pressure* extends *GROUND_freezeC_RichardsEq_seb* with a representation of soil mechanical processes. The new features are presented below in Sects. 2.1.1 and 2.1.2, as well as Table 1.

**Table 1.** Different model components, their implementation in the CryoGrid community model after Westermann et al. (2023) and the additions to the model code, presented in this study.

| Model component | Base class *GROUND_freezeC_RichardsEq_seb* | Additions in *GROUND_freezeC_RichardsEq_seb_pressure* |
|---|---|---|
| Upper boundary condition | Surface energy balance | - |
| Lower boundary condition | Geothermal heat flux | - |
| Subsurface heat transfer | Heat conduction and heat convection | - |
| Soil freezing characteristics | after Painter and Karra (2014) | - |
| Water flow in unsaturated conditions | Richard's equation | - |
| Water flow in saturated conditions | Gravity-driven | Additional water flow when excess pore water pressures occur |
| Lateral water transport | Overland flow | - |
| Stress conditions in the ground | - | Calculation of total and effective stresses |
| Compressibility of the soil column | - | Calculated from the compression curve |
| Excess ground ice | Defined in the initial conditions, melting of excess ice possible | Formation and melt of segregated ice |
| Sedimentation | - | Material is added with user-defined properties |

### 2.1.1 Soil mechanical processes

We build on established concepts for ground consolidation, following Dumais and Konrad (2018). This includes the calculation of (i) the total stress, (ii) the effective stress, (iii) the compaction of the soil and the water fluxes (Sect. 2.1.2), which are

updated in every time step. To do so, a set of additional state variables is necessary, which can be found in Table 2. A schematic illustration of the stresses acting on the soil column is depicted in Fig. 2.

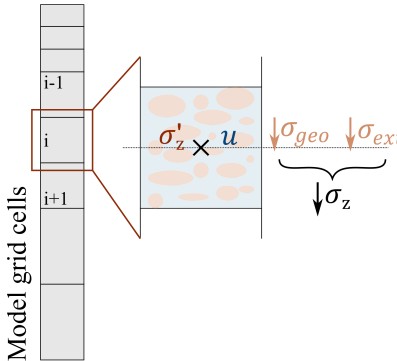

**Figure 2.** Illustration of the stresses acting on the soil column with geostatic stress $\sigma_{geo}$, external stress $\sigma_{ext}$ and pore water pressure $u$. Their relationship as well as the computation of the effective stress $\sigma'_z$ is described in Eq. (3) and Eq. (4).

(i) The total stress $\sigma_z$ [Pa] is the sum of both external stress $\sigma_{ext}$ [Pa] and the vertical geostatic stress $\sigma_{geo}$ [Pa]:

$$\sigma_z = \sigma_{ext} + \sigma_{geo}. \tag{2}$$

External stress is caused by additional loads on the ground surface such as buildings or roads and acts with the same value on all grid cells in the soil column. In contrast, the vertical geostatic stress is caused by the mass of the overburden soil and consequently increases with depth. In our model, we calculate the total stress at the midpoint of each grid cell. Therefore, we consider the weight of all grid cells above in addition to half of the weight of the grid cell itself. The geostatic stress can be determined by dividing the weight of the soil, $W$ [$\mathrm{kg\,m\,s^{-2}}$], by the area $A$ [$\mathrm{m^2}$]:

$$\sigma_{geo} = \frac{\sum_i W_i}{A} = \frac{\sum_i \Theta_i\, \rho_i\, g}{A} \tag{3}$$

with $\Theta_i$ [$\mathrm{m^3}$] being the volume, and $\rho_i$ [$\mathrm{kg\,m^{-3}}$] the density of a grid cell $i$. As grid cells consist of mineral, organic, water and ice constituents, the density is calculated as the arithmetic average of the constituent densities weighted by their volumetric fractions.

(ii) The effective stress $\sigma'_z$ [Pa] describes the stress acting on the soil matrix and is defined as the total stress $\sigma_z$ [Pa] minus the pore water pressure $u$ [Pa]:

$$\sigma'_z = \sigma_z - u. \tag{4}$$

To compute the pore water pressure, the buoyancy effect has to be accounted for. For each grid cell of the soil column, which is composed of one soil type (sand, silt, clay or organic), the saturation is calculated separately. When the soil is at or near saturation, the porewater results in a reduction of the stress on the soil matrix, as the density of each component is reduced

by the density of water. For saturated conditions, we assume that the buoyancy effect is fully effective. If the saturation is less than 50 %, we make the assumption that the total stress is carried exclusively by the soil matrix and the pore water pressure is set to zero. To facilitate a continuous transition of pore water pressure between saturated (100 %) and dry (< 50 %) conditions we apply a linear interpolation. While this is a coarse approximation, neglecting the increase in effective stress where the soil is saturated due to capillarity, it allows us to calculate the effective stress in the soil column avoiding step changes between the two regimes.

(iii) When the soil compresses or relaxes, the porosity changes, which must be equilibrated by either fluxes of air or water. In this work, we assume that air fluxes occur instantly, i.e. for unsaturated conditions, the soil is compacted to its equilibrium position after each time step. The compressibility of a soil can be expressed with a linear relationship between void ratio $e$ [-] (defined as the ratio of pore volume to the volume of mineral and organic matter) and the decadic logarithm of the effective stress $\sigma'_z$ [Pa] (Murthy, 2002) as illustrated in Fig. 3. With increasing effective stress, the void ratio decreases linearly, with steeper curves indicating a higher compressibility of the soil. The slope of the curve is given by the compression index $C_c$ [-], which can be expressed as

$$C_c = -\frac{e - e_0}{\log(\sigma'_z) - \log(\sigma'_0)} \tag{5}$$

with the initial void ratio $e_0$ [-] and the residual stress $\sigma'_0$ [Pa] staying constant during the simulation. The latter is defined as the effective stress, where no consolidation occurs upon thawing in undrained conditions, i.e. the effective stress that can be maintained by the soil skeleton (Nixon, 1973; Dumais and Konrad, 2018). The effective stress is calculated in Eq. (4), so that the void ratio can be computed from Eq. (5):

$$e = e_0 - C_c \log(\frac{\sigma'_z}{\sigma'_0}). \tag{6}$$

Knowing the void ratio $e$ [-] of the soil, the porosity $\phi$ [-] of a grid cell can be calculated by the following equation:

$$\phi = \frac{e}{1 + e} \tag{7}$$

With changing porosity, the volume of the affected grid cell has to be adapted. As the area $A$ [m$^2$] is fixed throughout the entire simulation, the volume change is reflected in the thickness of a grid cell $d$ [m]:

$$d = \frac{\Theta_m + \Theta_o}{(1 - \phi) A} \tag{8}$$

with $\Theta_m$ [m$^3$] and $\Theta_o$ [m$^3$] being the bulk volumetric content of the mineral and organic components, i.e. the absolute volume in a grid cell filled by the respective component. In *GROUND_freezeC_RichardsEq_seb*, bulk quantities as $\Theta_m$ and $\Theta_o$ are conveniently used as state variables (Westermann et al., 2023), so that the grid cell thickness $d$ for unsaturated grid cells can be updated with the porosity obtained from Eq. 7 (see Sect. 2.1.2 for the saturated case).

**Table 2.** State variables for soil mechanical properties

| State variable | Symbol | Unit |
|---|---|---|
| Total stress | $\sigma_z$ | Pa |
| External stress | $\sigma_{ext}$ | Pa |
| Vertical geostatic stress | $\sigma_{geo}$ | Pa |
| Effective stress | $\sigma'_z$ | Pa |
| Pore water pressure | $u$ | Pa |
| Initial void ratio | $e_0$ | - |
| Residual stress | $\sigma_0$ | Pa |
| Compression index | $C_c$ | - |

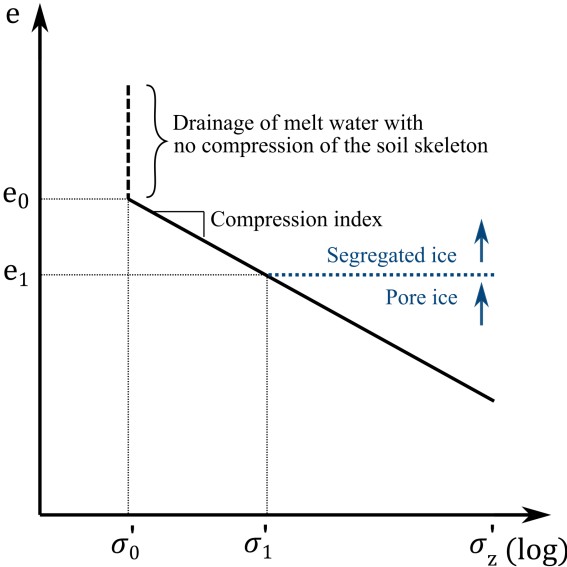

**Figure 3.** Relationship between void ratio $e$ and effective stress $\sigma'_z$ for thawing soils. If ice segregation results in void ratios larger than the initial void ratio $e_0$, melt water produced by thawing of such grid cells drains without compression of the soil skeleton, as the effective stress stays constant at the residual stress $\sigma'_0$. For increasing effective stress, the soil compresses according to the compression index $C_c$. The combination of void ratio and effective stress, which the soil has in a compressed state without excess pore water pressures under the applied overburden pressure, are indicated with $e_1$ and $\sigma'_1$. All ground ice exceeding the available pore space at this stress regime is defined as segregated ice.

### 2.1.2  Water fluxes

The model calculates fluxes of soil water in both unsaturated and saturated soil as shown in Fig. 4. Water fluxes in unfrozen and unsaturated soil are computed from gradients in (i) matric potential and (ii) gravitational potential based on the Richards equation in the same way as presented for the stratigraphy class *GROUND_freezeC_RichardsEq_seb* in Westermann et al. (2023) and described in Sect. 2.1. (i) Gradients in matric potential induce a vertical water flow between grid cells. This process is especially active in the upper soil layers, where rainfall, snow melt water and evaporation influence the soil water content.

For example, when upper grid cells loose soil water due to evaporation, the matric potentials of the affected grid cell decrease, leading to suction of soil water from the grid cells below and inducing upward water fluxes. (ii) The gravitational potential always leads to water fluxes from the top to the bottom of the ground column, e.g. the infiltration of rainfall or melt water. The highest value occurs in the uppermost grid cell and with progressing depth, the value decreases according to the thickness of the grid cells. For saturated conditions, the gravitational potential in the aquifer matches the hydrostatic potential, i.e. the value

at the top of the aquifer and as a result, no water fluxes occur within the aquifer due to the gravitational potential. It is important to highlight that water fluxes in unsaturated soil do not directly lead to a change in grid cell size in the presented model scheme, as changing water content are replaced by the same volume of air. However, the resulting changes in water content affect the stress level in the soil and therefore the soil mechanical responses.

If the soil is unfrozen and saturated, downward water fluxes occur in case unsaturated grid cells are present in deeper layers,

which attract soil water through both gravitational and matric potential similar to *GROUND_freezeC_RichardsEq_seb*. In case of an aquifer, where the entire soil column below is saturated, no water is exchanged between the saturated grid cells. In these conditions, the effective stress $\sigma_z'$ [Pa] on the soil skeleton can be calculated by reducing the total stress $\sigma_z$ [Pa] by the pore water pressure $u$ [Pa], as shown in Eq. 4.

When external loads are applied or sedimentation takes place, the additional stress is taken up by the soil water in a first

step, leading to excess pore water pressures $u_e$ [Pa]:

$$u_e = \sigma_z - \sigma_z',  \tag{9}$$

which results in water fluxes $j_{w,u_e}^v$ [m s$^{-1}$] away from the affected grid cell

$$j_{w,u_e}^v = -\frac{K_w}{\rho_w g}\left(\frac{\partial u_e}{\partial z}\right).  \tag{10}$$

leading to a reduction of the excess pore water pressure by consolidation. The water flux is added to the fluxes calculated based

on the Richards equation (Eq. 1). The consolidation continues until the excess pore water pressure reaches a value of zero, while at the same time the effective stress is increased. As the finite hydraulic conductivity of the soil controls the water fluxes, this process takes a certain amount of time. Therefore, saturated soils cannot immediately respond to changing stresses: Before the soil skeleton can take the additional stress and compress, the water has to flow out of the pores. The opposite effect takes place, when a formerly dry soil becomes saturated and the increased pore water pressures reduce the effective stress on the soil

skeleton.

After calculating the water fluxes in the soil column, the change in water content for each grid cell can be derived. For unsaturated conditions, changing water content is replaced by air. For saturated conditions, no air inflow is possible and changes in water content affect the grid cell size. The thickness of a grid cell $d$ [m] can be calculated for saturated conditions from the volume $\Theta$ [m$^3$] of the grid cell (being the volume of water, ice, mineral, organic) divided by the area $A$ [m$^2$]:

$$d = (\Theta_w + \Theta_i + \Theta_m + \Theta_o)/A \tag{11}$$

With the thickness of the grid cell $d$ being calculated for saturated conditions directly from the change in water content, we can invert Eq. 8 and Eq. 7 and solve Eq. 6 for the effective stress $\sigma_z'$ to update the excess pore water pressure with Eq. 9. A reduction/increase in water content results in a change in effective stress and thus in compression/swelling of the soil.

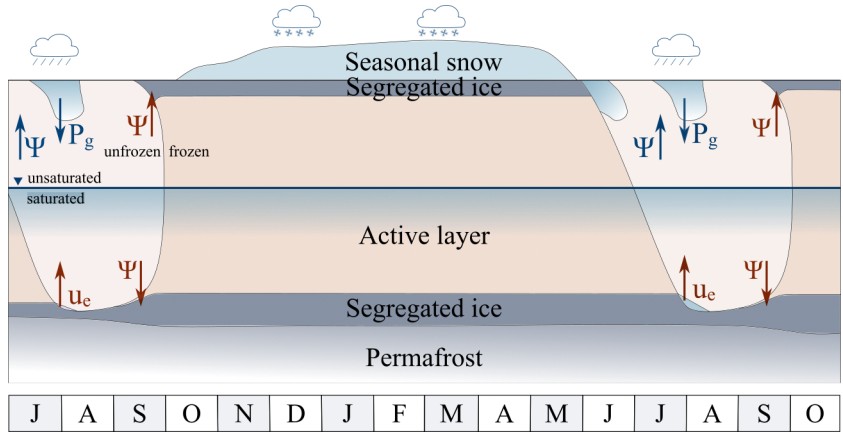

**Figure 4.** Illustration of the potentials resulting in water fluxes as calculated in the model (not to scale): During winter season, the entire soil column is frozen and no substantial water fluxes occur. When the upper soil layers are unfrozen during summer season, water fluxes in the unsaturated zone are controlled by the gravitational potential $P_g$ and the matric potential $\psi$. Rainwater or meltwater infiltrates from the top to deeper layers. Most important for ice segregation is the matric potential during fall refreezing (marked in red). In case the downward thawing from the surface during summer reaches (partly) the segregated ice, excess pore water pressures $u_e$ develops, leading to thaw consolidation.

### 2.1.3 Segregated ice

When the soil freezes, water fluxes are significantly smaller than in unfrozen conditions due to reduced liquid water contents and hydraulic conductivity (Burt and Williams, 1976; Horiguchi, 1983). However, the remaining soil water is still partly mobile. This is mainly driven by matric potentials, which reach considerably negative values for ground temperatures below zero degrees, resulting in an attraction of soil water towards the freezing front (Perfect and Williams, 1980; Williams and Smith, 1989). In contrast to the stratigraphy class *GROUND_freezeC_RichardsEq_seb*, the new model scheme enables water fluxes also into saturated grid cells. In this case, the grid cell gradually expands and the void ratio increases. When its void ratio reaches the initial void ratio, the entire weight of the overlying soil is compensated by the matric potential. Water fluxes into the grid cell continue, as long as (i) the difference in matric potential is large enough to compensate for the weight of the overlying

soil and (ii) the hydraulic conductivity of the freezing soil allows water fluxes into the affected grid cell. With increasing ice content in the grid cell, the hydraulic conductivity is strongly reduced, which effectively stops water redistribution and leads to a stable situation. Upon thawing, excess pore water pressure leads to water fluxes out of the affected grid cell and thaw consolidation takes place.

We highlight that the term "segregated ice" is defined differently within the framework of the CryoGrid community model than the term "excess ice" in previous versions of the model. Westermann et al. (2016) and Westermann et al. (2023) distinguish between an incompressible soil phase including pore water/ice, and an excess ice phase, which becomes mobile upon melting and thus leads to a shrinking of the grid cell. In contrast, the presented stratigraphy class in this study defines segregated ice as the ice volume exceeding the pore space, which the soil would have in a compressed state without excess pore water pressures under the applied overburden pressure (illustrated in Fig. 3). This implies that for a given soil a small additional ice content at great depth is be defined as segregated ice, while the same volumetric ice content could be present as regular pore ice at lower depths. We note that the new model scheme can only represent segregated ice and that the formation of other forms of excess ice such as wedge ice cannot be accounted for as they are associated with different processes during formation.

Segregated ice that is present at void ratios smaller than the initial void ratio is considered to be bound within the pore structure of the soil and the partitioning of ice and water thus follows the soil freezing characteristics of the given soil type (Westermann et al., 2023). In contrast, any further segregated ice exceeding the initial void ratio is assumed to be "free" water, which freezes at $T = 0\,^{\circ}\text{C}$.

### 2.1.4 Sedimentation

The model provides the option to include sedimentation by organic and mineral matter in the simulations (stratigraphy class *GROUND_freezeC_RichardsEqW_seb_pressure_sedimentation*). The user can define the sedimentation rate and the depth at which the material is added. In the current version, the added material is assumed to have the same composition as the already present soil at the time of sedimentation. However, this is not a principle limitation, and sedimentation with user-defined properties could be implemented in a straight-forward way. Furthermore, sedimentation only occurs for positive ground temperatures in the current model version. Water is not added during sedimentation, but the newly created pore space can be filled through water flow from neighboring grid cells or rain/snow melt (Sect. 2.1.2). When the grid cell exceeds a certain target thickness, the grid cell is split into two to maintain a set minimum resolution in the soil column. Intensive variables (e.g. temperature and porosity) are inherited for both grid cells, while extensive parameters (e.g. bulk mineral and organic content, Westermann et al., 2023) are divided between them while maintaining the proportions during the split.

### 2.2 Field sites and forcing data

To demonstrate the capabilities of the new model, we perform sensitivity studies (Sect. 2.4) for two different permafrost sites with strongly different climate conditions: Samoylov island (located in northern Siberia) represents a continental setting with cold continuous permafrost, while the Bayelva field site (located on Svalbard) is characterized by a maritime climate with warm, but still continuous permafrost (Fig. 5).

Samoylov island is located in northeastern Siberia in the Lena River delta (72°22' N, 126°28' E). Monitoring of soil temperatures and meteorological conditions has been performed at the research site since 1998 (Boike et al., 2013, 2019). Due to the geographic location, Arctic continental climate leads to mean annual air temperatures of below -12 °C. Summer air temperatures can reach up to 28 °C, while minimum air temperatures in winter season can fall below -45 °C. In summer, precipitation and evapotranspiration balance each other in Samoylov. Winter season is typically characterized by a thin snow cover with a mean end-of-winter thickness of 0.3 m (Boike et al., 2013; Gouttevin et al., 2018; Langer et al., 2011b). Samoylov is situated in continuous permafrost, which reaches depths between 400 and 600 m (Grigoriev, 1960). East of the research station, the landscape is shaped by polygonal tundra with ice wedge polygons dominating the upper meters of the fine-grained and organic-rich soils. There are both low- and high-centered polygons, and the local microtopography is largely structured by sequences of polygon troughs, rims, and centers (Boike et al., 2013). The observed ground temperature at a depth of 20.75 m increased from -9.0 °C in 2007 to -7.9 °C in 2016 (GTN-P, 2018). With these records, the field site shows one of the strongest warming of permafrost within 123 globally distributed boreholes (Biskaborn et al., 2019). The ground temperature for model validation was taken beneath a polygon center close to the research station (Boike et al., 2013). We use the same forcing data set as Westermann et al. (2016), Nitzbon et al. (2019) and Nitzbon et al. (2020) for the long-term thaw susceptibility runs in that study. The climatic forcing between 1960 and 2012 is derived from the reanalysis product CRU-NCEP (combining data from the Climate Research Unit and National Centers for Environmental Prediction, see Kalnay et al., 1996; Harris et al., 2014), downscaled for Samoylov with in situ data from the automatic weather station. After 2012, the forcing data is taken from the Community Climate System Model CCSM4 outputs simulated under the Representative Concentration Pathway (RCP) 8.5 scenario (Meehl et al., 2012). To better match site measurements, statistical downscaling is performed for incoming long-wave radiation, air temperature and absolute humidity, using linear regression between measurements and CRU-NCEP and CCSM4. Comparing the reference period 1961-1990 to 2080-2100, the forcing data set shows the following trends: An increase in air temperature from -16.7 °C to -8.1 °C, an increase in longwave radiation from 214 to 249 $W\,m^{-2}$, a slight decrease in short-wave radiation from 105 to 101 $W\,m^{-2}$, a pronounced increase in rainfall from 157 to 218 mm and an increase in snowfall from 133 to 167 mm. A detailed description of the forcing data set can be found in Westermann et al. (2016).

In addition, we use the data set from Bayelva, Svalbard (78°55' N,11°50' E). In contrast to the continental climate in Samoylov, Bayelva represents a maritime setting with smaller seasonal fluctuations in air temperature. The field site conditions are controlled by the West Spitsbergen Current leading to a relatively mild climate for that latitude. Measured annual mean air temperature in Ny-Ålesund was -3.1 °C between 2010 and 2021, with 2010 having the lowest (-3.6 °C) and 2017 the highest (-2.8 °C) value (Norwegian Meteorological Institute, 2022a). Mean annual precipitation was recorded to be 481 mm between 2000 and 2021 (Norwegian Meteorological Institute, 2022b). It falls predominantly as snow between September and May and the maximum snow depth typically varies between 0.65 and 1.4 m (Boike et al., 2018). The field site is characterized by hilly tundra with soils featuring silty clay to sandy silt with some larger stones and organic carbon concentration being highest in about 1 m depth (> 6 % weight) (Boike et al., 2008, 2018; Westermann et al., 2009). Observed ground temperatures at a depth of 9 m varied between -3.0 and -2.6 °C during the period of 2007 to 2016 (GTN-P, 2018) and thus show relatively warm permafrost. We use the same model forcing as Schmidt et al. (2021) for the presented long-term runs in that study.

Between 1980 and 2019, the model forcing is derived from ERA-Interim reanalysis, while the future forcing is based on CMIP5 projections (RCP 8.5) of CCSM4 using an anomaly approach. For a detailed description of the forcing data set see Schmidt et al. (2021). Validation performed by Westermann et al. (2023) with different stratigraphy classes of the Cryogrid community model shows that CryoGrid can capture the ground temperatures at the Bayelva site well.

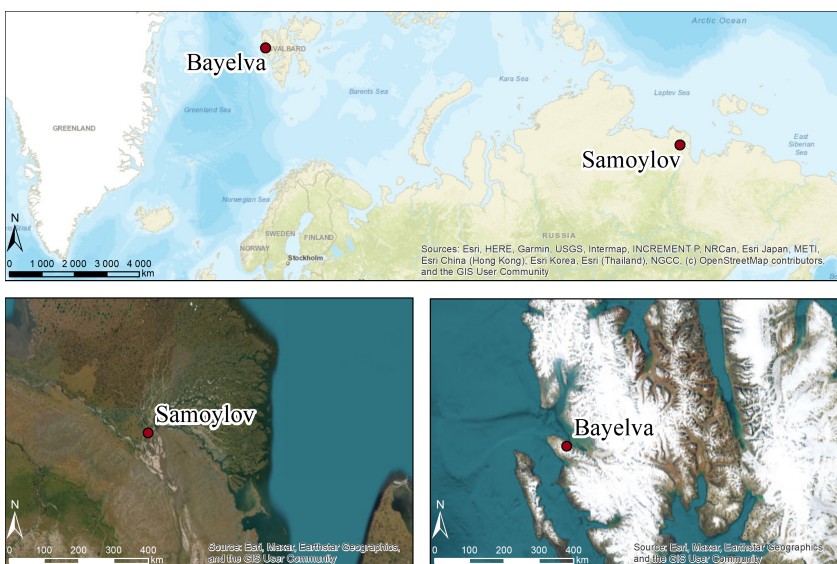

**Figure 5.** Location of Samoylov island in the Lena River delta in northeastern Siberia and of the Bayelva climate station on Svalbard. The aerial images of the surroundings of the field sites are shown in the lower left for Samoylov island and on the lower right of the Bayelva climate station.

## 2.3 Model parameters and initialization

We use a model domain reaching from the surface to 100 m depth, which is described by a stack of two different stratigraphy classes. Below 9 m depth, soil mechanical processes are not accounted for due to frozen conditions and high total stress during the entire simulation. Therefore, we apply the stratigraphy class *GROUND_freeW_seb*, which operates without soil mechanical processes and water balance, as described in Westermann et al. (2023). Between the ground surface and 9 m depth, we use the new stratigraphy class *GROUND_freezeC_RichardsEq_seb_pressure* as presented in this study. We chose the snow class *SNOW_crocus2_bucketW_seb*, based in the Crocus snow model (Vionnet et al., 2012), to represent seasonal snow cover with parameters adapted as in Westermann et al. (2023).

For model validation on Samoylov island, we set a soil stratigraphy as described for the center of a low-centred polygon in Holocene deposits in Samoylov in Nitzbon et al. (2020) and Westermann et al. (2016) (multi-layer stratigraphy in Table 3). In contrast to these studies, we do not assume any excess ice in the beginning of the simulation as it is inherently generated by the new model scheme. To represent the entire system of polygonal tundra, laterally coupled simulations (e.g. Aas et al., 2019; Nitzbon et al., 2019) would be required, but a one-dimensional soil column is sufficient for the purpose of this study.

To analyze the performance and sensitivity of our model for Bayelva and Samoylov, we used simplified stratigraphies to provide standardized and comparable model setups. Hereby, we use the same soil properties in the upper 0.15 m as in the multi-layer stratigraphy from Samoylov Island (see above) to account for the insulating effect of the organic-rich moss layer on top. The poorly decomposed organic material features coarse pores with unknown values for the van Genuchten parameters $n$ and $\alpha$. Therefore, it is phenomenologically set to the properties of coarse-grained (sandy) soil, which has a broadly similar retention characteristic. Between 0.15 and 9 m depth, we set homogeneous soil properties of one soil type, distinguishing between sand, silt, clay and peat. They feature different characteristics for volumetric fractions of mineral and organic content, saturation, initial void ratio, residual stress, compression index, permeability, $\alpha$ coefficient and $n$ coefficient. The chosen values for the different settings can be found in Table 3. The volumetric fractions of the mineral and organic content are taken from Westermann et al. (2016). The initial void ratio is calculated from the corresponding porosity. The soil is assumed to be fully saturated at the beginning of the simulations, only the surface layer has a saturation of 50 %. As the residual stress is unknown, we choose a value of 1000 Pa for all soils, except for organic-rich layers, which we set to 100 Pa, so that the corresponding thawed void ratio stays meaningful and realistic. A comparable range in values of residual stress has been applied by Dumais and Konrad (2018) and Dumais (2019). We estimate the compression index based on literature value (Gudehus, 1981), as well as the Van Genuchten parameters $\alpha$ and $n$ (Dall'Amico et al., 2011; Gnatowski et al., 2010; Van Genuchten, 1980). It should be noted that the mechanical and hydrological parameters are soil type dependent, but are adjusted independently in the model setup.

Other model parameters are set in accordance with previous model studies in the same areas. During the snow-free period, the albedo of the surface is set to 0.2 and an aerodynamic roughness length of 0.001 m is assumed for the ground (Westermann et al., 2016). We use the snow class based on the Crocus snow model (Vionnet et al., 2012; Zweigel et al., 2021), but increase the maximum wind slab density to 500 $kg\,m^{-3}$, in line with Barrere et al. (2017) and Royer et al. (2021). Furthermore, we define a maximum snow depth to account for snow ablation by wind drift as stated in Sect. 2.4 (Westermann et al., 2016).

For Samoylov, we used the same initial ground temperatures as Nitzbon et al. (2019), as we used the same forcing data as this study: 0 m depth: 0.0 °C; 2 m: -2.0 °C; 5 m: -7.0 °C; 10 m: -9.0 °C; 25 m: -9.0 °C; 100 m: -8.0 °C; 1100 m: +10.2 °C. These values are based on borehole data from 2006 and a steady-state temperature profile corresponding to a geothermal heat flux of 50 $mW\,m^{-2}$ (Langer et al., 2013). The initial ground temperatures for Bayelva were determined based on the 100 years spin-up ground temperatures, which are in line with Westermann et al. (2023): 0 m depth: 5.0 °C; 0.6 m: 0.0 °C; 1 m: -1.0 °C; 2 m: -2.0 °C; 10 m: -7.0 °C; 100 m: 1.0 °C. Since we conduct simulations on at least centennial timescales and the temperature profile in the uppermost meters becomes independent of the initialization after a few years, the initial temperatures do not affect simulation results. Furthermore, the soil column in the uppermost 9 m is compressed according to the mass of the overlying layers before the start of the simulations. For this pre-compaction, we assume an unfrozen conditions with the saturation given in Table 3.

The initial vertical resolution of the grid cells increases stepwise from the surface to greater depths (0-1 m depth: 0.05 m; 1-5 m depth: 0.1 m; 5-10 m depth: 0.2 m; 10-20 m depth: 0.5 m; 20-50 m depth: 1 m; 50-100 m depth: 5 m). The fine resolution in the top layers allows us to analyze the ground temperatures in detail in the upper soil layers.

**Table 3.** Stratigraphies used for the model scenarios in this study in the upper 9 m of the model domain. While the multi-layer stratigraphy represents a soil column as described in Nitzbon et al. (2020) and Westermann et al. (2016) for Samoylov island, the simplified stratigraphies consist of only one soil type throughout the entire soil column below the organic-rich layer at the top. Below a depth of 9 m, we define a base layer with a volumetric mineral content of 0.7 and a volumetric ice content of 0.3, which is assumed to be static and does not evolve due to soil compression. $\theta_m$: volumetric fraction of mineral content; $\theta_o$: volumetric fraction of organic content; $S$: saturation; $e_0$: initial void ratio before compaction; $\sigma_0$: residual stress; $C_c$: compression index; $k_w$: permeability; $\alpha$: alpha coefficient; $n$: n coefficient. We assume no residual water.

| Depth [m] | $\theta_m$ [-] | $\theta_o$ [-] | $S$ [-] | $e_0$ [-] | $\sigma_0$ [Pa] | $C_c$ [-] | $k_w$ [m$^2$] | $\alpha$ [m$^{-1}$] | $n$ [-] |
|---|---|---|---|---|---|---|---|---|---|
| Multi-layer | | | | | | | | | |
| 0 - 0.15 | 0.1 | 0.15 | 0.5 | 3 | 100 | 1 | $10^{-12}$ | 4.06 | 2.00 |
| 0.15 - 0.90 | 0.3 | 0.05 | 1 | 1.86 | 1000 | 0.2 | $10^{-13}$ | 0.65 | 1.70 |
| 0.90 - 9.00 | 0.55 | 0.05 | 1 | 0.67 | 1000 | 0.01 | $10^{-11}$ | 4.06 | 2.00 |
| Simplified sand | | | | | | | | | |
| 0 - 0.15 | 0.1 | 0.15 | 0.5 | 3 | 100 | 1 | $10^{-12}$ | 4.06 | 2.00 |
| 0.15 - 9.00 | 0.6 | 0 | 1 | 0.67 | 1000 | 0.01 | $10^{-11}$ | 4.06 | 2.00 |
| Simplified silt | | | | | | | | | |
| 0 - 0.15 | 0.1 | 0.15 | 0.5 | 3 | 100 | 1 | $10^{-12}$ | 4.06 | 2.00 |
| 0.15 - 9.00 | 0.55 | 0 | 1 | 0.82 | 1000 | 0.2 | $10^{-13}$ | 0.65 | 1.70 |
| Simplified clay | | | | | | | | | |
| 0 - 0.15 | 0.1 | 0.15 | 0.5 | 3 | 100 | 1 | $10^{-12}$ | 4.06 | 2.00 |
| 0.15 - 9.00 | 0.45 | 0 | 1 | 1.22 | 1000 | 0.5 | $10^{-14}$ | 1.49 | 1.25 |
| Simplified peat | | | | | | | | | |
| 0 - 0.15 | 0.1 | 0.15 | 0.5 | 3 | 100 | 1 | $10^{-12}$ | 4.06 | 2.00 |
| 0.15 - 9.00 | 0 | 0.2 | 1 | 4 | 100 | 1 | $10^{-12}$ | 2.31 | 1.29 |

Sedimentation is assigned to the grid cell below the organic-rich surface layer to assume a constant thickness of the layer influenced by vegetation. Consequently, the organic-rich surface layer is lifted with sedimentation, but stays unchanged regarding its soil properties. We set the sedimentation rate to $2\,\mathrm{mm\,a^{-1}}$, active only for positive ground temperatures conditions, which corresponds to an effective sedimentation rate of approximately $0.55\,\mathrm{mm\,a^{-1}}$. Furthermore, we doubled and tripled the sedimentation rate in two model scenarios (see Sect. 2.4), resulting in effective sedimentation rates of $1.1\,\mathrm{mm\,a^{-1}}$ and

$1.7 \, \mathrm{mm \, a^{-1}}$, respectively. While this is a simplified case, not trying to mimic a specific sedimentation process, it allows us to analyze the effect of material deposition on the accumulation and thaw of segregated ice.

## 2.4 Model scenarios

We set up different model scenarios by varying both forcing data and model parameters (Table 4). To validate our model for Samoylov island, we run a multi-layer stratigraphy based on observations (Table 3) for comparison with measurement data (*S-val*). All other scenarios are based on a simplified stratigraphy and are designed to analyze different influencing factors: (i) soil water contents and ground temperatures, (ii) soil type (sand, silt, clay and peat), (iii) external loads and (iv) sedimentation.

(i) We use two different forcing data sets to simulate conditions of permafrost in a cold environment (Samoylov, Siberia, model run *S-clay*) and permafrost closer to the thaw threshold (Bayelva, Svalbard, model run *B-clay*). We emphasize that the ground stratigraphy designed for Samoylov is not representative for the Bayelva field site, so that modelled ground temperatures cannot be compared to observations. All model scenarios that are forced by the Samoylov forcing include a spin-up by repeating ten times the years 1960 to 1969, while the period 1980-1989 is employed for Bayelva. Furthermore, we include one model run with the Samoylov forcing with rainfall reduced to 50 % to simulate drier conditions (*S-clay-rain50*). The spin-up of 100 years allows for a stable ground temperature profile, but ice segregation can still continue after the spin-up period. (ii) A comparison of the model runs *S-sand*, *S-silt*, *S-clay* and *S-peat* show the effect of different soil types on ice segregation and thaw consolidation. Hereby, we change the soil type in the simplified stratigraphy by adapting the mineral and organic content, the initial porosity, the residual stress, the compression index as well as $\alpha$ and $n$. In contrast to the other soil types, peat is characterized by a large porosity, low density and large water and organic matter contents. (iii) Further model scenarios include an external load of 5 kPa during the entire simulation (*S-clay-load5*) and from 1980, when the active layer thickness increases (*S-clay-load5-1980*). It corresponds to an about 40 cm thick gravel pad and allows us to discuss the influence of external loads both on ice segregation and on thaw consolidation. The load is applied instantaneously and does not change the surface properties (as a real gravel pad would certainly do), so that energy and water transfer occur as without the external load. (iv) Model scenarios *S-clay-sed100*, *S-clay-sed500*, *S-clay-sed1000* and *B-clay-sed1000* include sedimentation with a rate of $2 \, \mathrm{mm \, a^{-1}}$ over spin-up periods of 100, 500 and 1000 years, respectively. In addition, we performed two model scenarios with increased sedimentation rates for Samoylov (*S-clay-sed350-3x*) and Bayelva (*B-clay-sed370-2x*), where we reduced the spin-up period to 350 and 370 years to achieve similar total material deposition as in the 1000 year sedimentation runs.

For Samoylov, we set a threshold for the snow depth to account for observed snow ablation due to wind drift following the approach of Westermann et al. (2016). As the snow in the polygon centers is protected by the surrounding rims, we set a value of 0.45 m, which is in line with measured snow depths in polygons centers on Samoylov island (Boike et al., 2013). For validation, we use temperature measurements in a polygon center. However, these measurements are not conducted in the middle of Samoylov island, but close to the edge of the island where the wind drift is stronger and the height of the rims above the polygon center is reduced. Therefore, we reduce the maximum snow depth for the validation run to 0.20 m. For the Bayelva field site, we do not set a limit for the snow depth as the accumulation of the snow resulting from the forcing data generally represents local conditions well (Westermann et al., 2023).

Surface water loss by overland flow are based on the Gauckler-Manning equation (Westermann et al., 2023). As the terrain at the field site on Samoylov island is flat, we set a small gradient of $0.1\,\mathrm{m\,km^{-1}}$. In contrast, the Bayelva field site is situated on gently sloping terrain, so that we increased the gradient to $1\,\mathrm{m\,km^{-1}}$.

**Table 4.** Model scenarios

| scenario name | spin-up | forcing | stratigraphy | load | sedimentation | max. snow depth |
|---|---|---|---|---|---|---|
| *S-val* | 10 x 1960-1969 | Samoylov | multi-layer | - | - | 0.20 m |
| *S-clay* | 10 x 1960-1969 | Samoylov | clay | - | - | 0.45 m |
| *S-clay-rain50* | 10 x 1960-1969 | Samoylov (50 % rain) | clay | - | - | 0.45 m |
| *B-clay* | 12 x 1980-1989 | Bayelva | clay | - | - | - |
| *S-sand* | 10 x 1960-1969 | Samoylov | sand | - | - | 0.45 m |
| *S-silt* | 10 x 1960-1969 | Samoylov | silt | - | - | 0.45 m |
| *S-peat* | 10 x 1960-1969 | Samoylov | peat | - | - | 0.45 m |
| *S-clay-load5* | 10 x 1960-1969 | Samoylov | clay | 5 kPa | - | 0.45 m |
| *S-clay-load5-1980* | 10 x 1960-1969 | Samoylov | clay | 5 kPa (1980) | - | 0.45 m |
| *S-clay-sed100* | 10 x 1960-1969 | Samoylov | clay | - | $2\,\mathrm{mm\,a^{-1}}$ | 0.45 m |
| *S-clay-sed500* | 50 x 1960-1969 | Samoylov | clay | - | $2\,\mathrm{mm\,a^{-1}}$ | 0.45 m |
| *S-clay-sed1000* | 100 x 1960-1969 | Samoylov | clay | - | $2\,\mathrm{mm\,a^{-1}}$ | 0.45 m |
| *S-clay-sed350-3x* | 35 x 1960-1969 | Samoylov | clay | - | $6\,\mathrm{mm\,a^{-1}}$ | 0.45 m |
| *B-clay-sed1000* | 100 x 1980-1989 | Bayelva | clay | - | $2\,\mathrm{mm\,a^{-1}}$ | - |
| *B-clay-sed370-2x* | 37 x 1980-1989 | Bayelva | clay | - | $4\,\mathrm{mm\,a^{-1}}$ | - |

## 3   Results

In the following, we present results of the model scenarios described in Sect. 2.4 and summarized in Table 4. To ensure comparability between the different sites and years, results are always provided at August 31.

### 3.1   Model validation

For the Bayelva site, a detailed assessment of the performance of the CryoGrid community model is presented in Westermann et al. (2023), suggesting that the model can capture the key properties of the ground thermal regime. For the Samoylov island site, we perform model validation focusing on polygon centers where segregated ice is normally found. In detail, we compare the validation run *S-val* designed to represent the typical ground stratigraphy of polygon centers at the site (Sects. 2.3, 2.4) to published in situ temperature data from 0.05 m and 0.40 m depth at a polygon center near the northern shore of Samoylov island (Boike et al., 2013; Langer et al., 2011a, b; Westermann et al., 2016). Figure 6 shows the mean daily ground temperatures

for both measured and modelled data. The ground temperatures are well reproduced by the model and the seasonality of the measured signal is captured well. However, modelled ground temperatures are slightly too warm during summer in both 0.05 m and 0.40 m depth. In fall, modelled ground temperatures decrease earlier than the measurements and the zero curtain is underestimated. The same effect occurs in the simulations made by Westermann et al. (2016). A possible explanation is a delayed start of the snow season in the forcing data (Westermann et al., 2016) as the onset of snow cover in the applied forcing

data (based on reanalysis data) can differ by more than a month to the start of the snow season detected by automatic cameras in the field (Boike et al., 2013). This is explained by uncertainties in distinguishing between snow- and rainfall in the reanalysis product (Westermann et al., 2016) which likely causes the cold bias in our simulations during fall.

     The results of the validation run for Samoylov and the validation performed by Westermann et al. (2023) for Bayelva suggest that the model system including CryoGrid and model forcing can capture the main characteristics of the ground thermal regime

at the two sites. In the following, we use this model setup to perform a sensitivity analysis for ice segregation and thaw consolidation with the model scenarios described in Sect. 2.4.

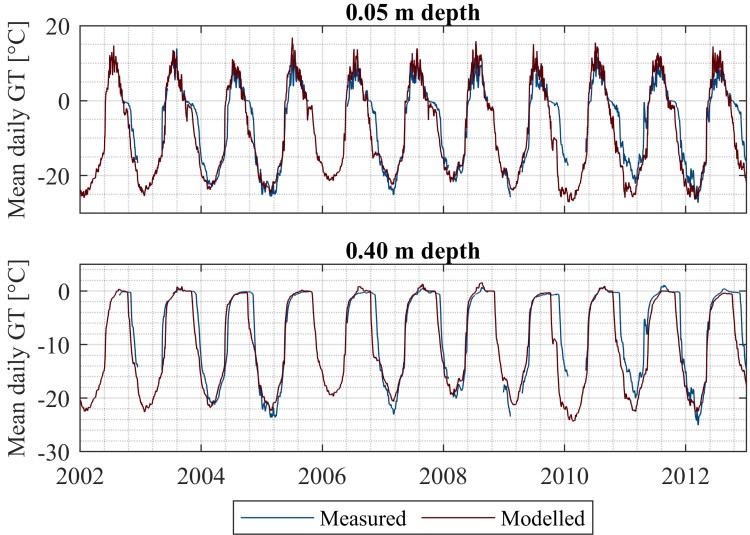

**Figure 6.** Measured and modelled ground temperatures at a depth of 0.05 m and 0.40 m. Measured temperature data is taken in a polygon center at the Samoylov field site. Results of the model scenario *S-val* can capture the measured ground temperatures in the two different depths.

## 3.2   Formation of segregated ice and thaw consolidation

The model run *S-clay* using a simplified stratigraphy of soil type clay shows the formation of segregated ice in the relatively cold period until the 1980s, as well as thaw consolidation during the past decades and the future forcing. During the spin-up,

mean annual ground temperatures $MAGT$ are modelled with values around -9 °C at a depth of 10 m depth. The relatively

cold period is then followed by a significant rise of $MAGT$. At the end of the 21st century, $MAGT$ at 10 m depth reach temperatures close to 0 °C (Fig. 7). The increasing ground temperatures are also reflected in the depth of the permafrost table. While the active layer is less than 0.70 m during the spin-up, a deepening can be observed since the 1980s (Fig. 7). In the early 2080s, the seasonally frozen ground does not reach the permafrost table anymore. A continuously unfrozen zone with positive ground temperatures throughout the year (talik) develops, indicating the degradation of the permafrost. The modelled changes in annual ground temperatures are in the same range as simulation results for the Samoylov field site in Westermann et al. (2016), as pointed out by the following comparison of approximate ground temperatures in 10 m depth (data of Westermann et al. (2016) shown in brackets): -9 °C (-10 °C) during the spin-up, -5 °C (-5 °C) in 2040 and -1 °C (-1 °C) in 2090.

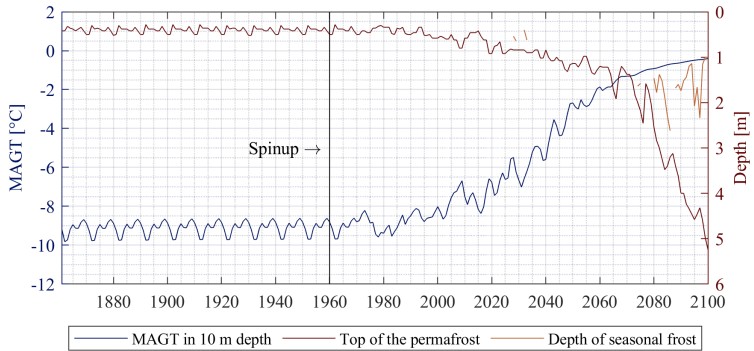

**Figure 7.** Mean annual ground temperatures $MAGT$ in 10 m depth, depth of the top of the permafrost and the seasonal frost in the model scenario *S-clay*. The spin-up covers the years 1860 to 1960. $MAGT$ warm from around -9 °C until the 1980s to values close to the thaw threshold at the end of the 21st century. The depth of the permafrost table deepens from around 0.7 m to 5 m, so that the seasonal frost doesn't reach the same depth towards the end of the century and a talik develops.

The influence of the changing climatic conditions is also apparent in the formation and thawing of the ground ice (Fig. 8). Ground ice accumulates during fall refreezing at the freeze front and the upper part of the permafrost due to differences in matric potential, which draw liquid water from the active layer into the frozen ground. While the model run is initialized with saturated conditions below 15 cm depth, the soil water content adapts to 70-80 % saturation in the active layer in the first year and remains at these values during spin-up. The process of ice segregation continues as long as thermal conditions allow a mobilisation of soil water, dependent on the freezing characteristics of the soil. The increased ice content is preserved at depths to which the active layer of the following years does not reach. As a consequence, *S-clay* builds up 0.034 m of segregated ice until the 1980s. Simultaneously, the ground surface heaves over the years by 0.044 m. Hence, most of the ground heave can be explained by ice segregation, while only a small amount is related to soil mechanical processes in the active layer, i.e. swelling and shrinking of the soil due to changes in soil water content caused by precipitation and evapotranspiration. In the following decades, warmer climate conditions lead to a deepening of the active layer. The segregated ice formed during spin-up starts to melt from the top to the bottom and thaw consolidation takes place. The resulting melt water flows upwards in the active layer

and the soil subsides again by 0.082 m until 2100. As only 0.034 m can be explained by melting of the segregated ice, the remaining subsidence is due to a drying of the soil column for the deeper soil layers at the end of the century (years 2080 to 2100, see Fig. S1 in Suppl. S2), increasing the effective stress.

When the active layer stabilizes for several years at a specific depth, new segregated ice is formed at the upper part of the permafrost (Fig. 8), which can lead to temporary surface heave even though the ground surface elevation decreases in the long term. However, the changes in ground surface elevation are too large to be only explained by new ice segregation, especially for the period between 2022 and 2100 (Fig. 9). Highly saturated conditions in the upper soil layers due to temporarily increased precipitation during the time period 2020-2100 (Fig. S1 in Suppl. S2, e.g. years 2070 to 2080) influencing the soil behaviour: The buoyancy effect of the water decreases the effective stress on the soil structure, which results in an temporarily increased void ratio and thus soil swelling.

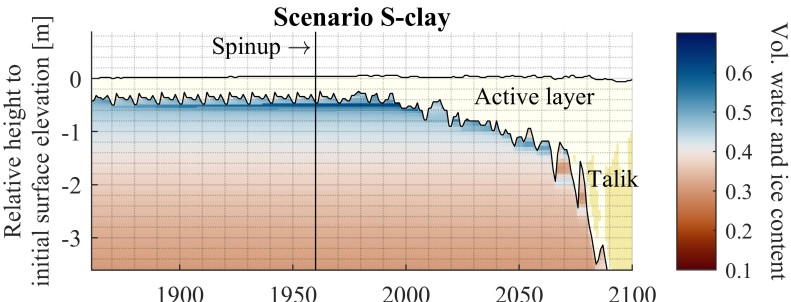

**Figure 8.** Sum of volumetric water and ice content in the permafrost on the reference date August 31 for the model run *S-clay*. Dark blue colors indicate an increased volumetric water and ice content at the top of the permafrost, where ice segregation takes place. Values in the thawed layer are not displayed. The spin-up covers the years 1860 to 1960. The sum of volumetric water and ice content is shown as soil water can still occur below freezing temperatures dependent on the soil type. For scenario setup see Table 4.

### 3.3 Sensitivity towards soil water content and ground temperatures

Model simulations with different forcing data sets suggest that ice segregation is highly dependent on the climatic conditions, which can lead to different soil water contents and ground temperatures. Figure 9 shows the formation of segregated ice and changes in surface elevation for the model runs *S-clay* (Samoylov forcing data), *S-clay-rain50* (Samoylov forcing data with 50 % rainfall) and *B-clay* (Bayelva forcing data). The reduction of rainfall in *S-clay-rain50* leads to less segregated ice during the spin-up on average with 0.025 m compared to 0.034 m in *S-clay* and consequently less ground heave. The reduced rainfall results in drier conditions in the active layer (Fig. S2 in Suppl. S2), so that less water is available for the formation of segregated ice. Furthermore, the increased negative matric potential in the drier active layer counteracts the matric potential responsible for ice segregation. Furthermore, drier conditions lead to less swelling in the active layer, so that fluctuations in surface elevation, especially during the time period 2020-2100, are dampened (Sect. 3.2). In model scenario *S-clay-rain50*, the permafrost does not degrade to the end of the 21st century and no talik develops. This can most likely be explained with a lower thermal

conductivity of the soil under drier conditions (increasing from 1.04 $\mathrm{W\,m^{-1}K^{-1}}$ (66 % saturation) to 1.67 $\mathrm{W\,m^{-1}K^{-1}}$ (100 % saturation), but other effects such melting of ground ice might play a role.

When the same model set-up is run with forcing data from Bayelva (*B-clay*), which represents warmer permafrost under
480 moist conditions, we obtain a comparable formation of segregated ice with a slightly lower maximum of 0.028 m in the 1980s compared to 0.034 m with forcing data from Samoylov (*S-clay*). As the results discussed above have shown that moist conditions lead to more segregated ice, one would expect higher values for *B-clay*. However, the maritime setting results in smaller temperature gradients in the ground during fall refreezing (Fig. S4 in Suppl. S2), which seem to compensate the effect of the soil moisture. Despite less ice segregation, the ground heave is more pronounced in Bayelva during the spin-up, due
to wetter conditions and a deeper permafrost table, resulting in stronger soil swelling. During the time period 2020 to 2100, *B-clay* builds up more segregated ice in deeper soil layers. This can be explained by a slower increase of the active layer thickness compared to Samoylov island, so that the permafrost table stabilizes for longer time periods, enabling new formation of segregated ice. As a consequence, less ground subsidence takes place for the Bayelva run. Varying the climatic forcing shows, that the model results depend on both soil moisture and temperature gradients in the ground, controlling the water
migration towards the freezing front (Burt and Williams, 1976; Perfect and Williams, 1980).

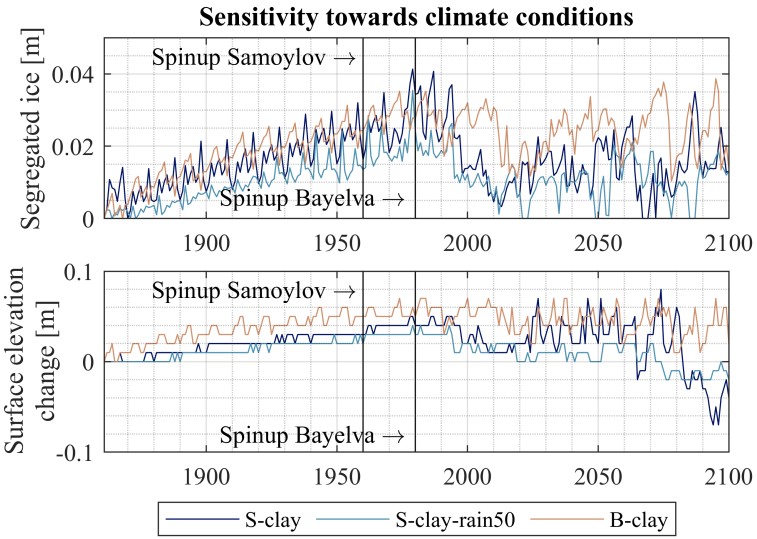

**Figure 9.** Column-accumulated segregated ice and surface elevation change on the reference date August 31 of each year in the simulation, for the model scenarios *S-clay*, *S-clay-rain50* and *B-clay*. Drier conditions lead to less formation of segregated ice and thus less ground heave. The moist conditions in *B-clay* are compensated by smaller temperature gradients in the ground so that similar segregation ice contents are formed during model spin-up.

## 3.4 Sensitivity towards soil type

The soil type has a strong influence on ice segregation and thaw consolidation (Fig. 10). The formation of segregated ice in the model run with the sand layer (*S-sand*) is negligible and no significant ground heave occurs. Yet, the other soil types form segregated ice and reveal a distinct change in ground surface position. The model run with the silt stratigraphy (*S-silt*) builds up a maximum of 0.015 m segregated ice until the 1980s and heaves by 0.018 m. The reference run with the clay stratigraphy (*S-clay*) has a slightly stronger reaction with a build-up of segregated ice of 0.034 m and a ground heave of 0.044 m (Sect. 3.2). In contrast to these reactions on a centimeter scale, the model run with the peat stratigraphy (*S-peat*) yields much more ice-rich ground, with a maximum of accumulated segregated ice of 0.784 m and ground heaving by by 0.815 m (Fig. 11). Here, the ground heave is predominantly caused by ice segregation, which is especially effective for peat due to large differences in matric potential and small vertical geostatic stress due to the low density of the peat, while soil swelling accounts for only a small part.

Upon active layer deepening, the segregated ice melts. The results show that a larger portion of the segregated ice melts between 1990 and 2000 in the silt and clay stratigraphy, while this happens between 2000 and 2010 for peat. A possible explanation for this effect is the lower thermal conductivity of the organic phase ($0.25 \ \mathrm{W \, m^{-1} \, K^{-1}}$) compared to the mineral phase ($3.0 \ \mathrm{W \, m^{-1} \, K^{-1}}$) as well as the higher amount of segregated ice that consumes energy upon melting. In the future projection, ice segregation occurs in deeper soil layers (at the top of the permafrost) even though it is typically less intense due to higher total stress and smaller temperature gradients. Therefore, segregated ice does not disappear completely for model runs *S-silt*, *S-clay* and *S-peat*, but potentially slows permafrost degradation as energy is needed for melting of the newly formed segregated ice in deeper soil layers. Soil swelling and shrinking influences the surface elevation, especially during the time period 2020-2100 (Sect. 3.2), but both model scenarios *S-clay* and *S-silt* result in a net subsidence of the ground surface.

## 3.5 Sensitivity towards external ground loading

Applying an external load influences the formation and thaw of segregated ice (Konrad, 1983, Fig. 12). The model run with an external load from the beginning of the simulation (*S-clay-load5*) compresses the entire soil column during initialization by 0.231 m and shows 30 % less segregated ice and less variation during the spin-up compared to the reference run *S-clay*. It reaches a maximum in the 1980s with approximately 0.025 m of segregated ice compared to 0.034 m in the reference run *S-clay*. In the following decades, both simulations follow a comparable curve when thawing. The model run *S-clay-load5-1980* applies an external load when the segregated ice is at its maximum in 1980. The ground settles within the first summer period by 0.101 m, which is less than the compression during initialization in model run *S-clay-load5*, as only the active layer is compressed, while the frozen ground below can bear the additional load. As the active layer is saturated with 70-80 %, the compression takes places immediately as described in Sect. 2.1.1 for unsaturated conditions. With the deepening of the active layer, the soil column compresses further. In addition, the compression leads to a slightly enhanced deepening of the active layer, which implicates the development of a talik in the 2060s instead of in the early 2080s as in model run *S-clay* and no segregated ice is in place at the end of the 21st century.

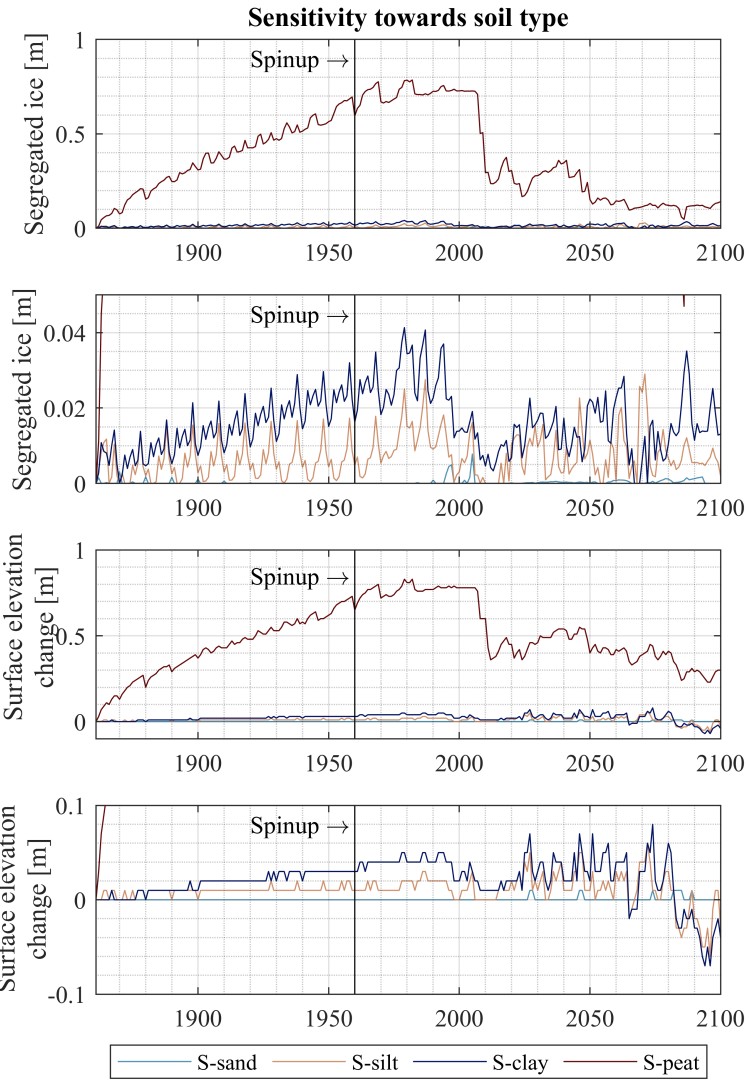

**Figure 10.** Column-accumulated segregated ice and surface elevation changes on the reference date August 31 for the model scenarios *S-sand*, *S-silt*, *S-clay* and *S-peat*. With decreasing particle diameter, the soil builds up more segregated ice under equal conditions. Peat soils form substantially more ground ice than mineral soils.

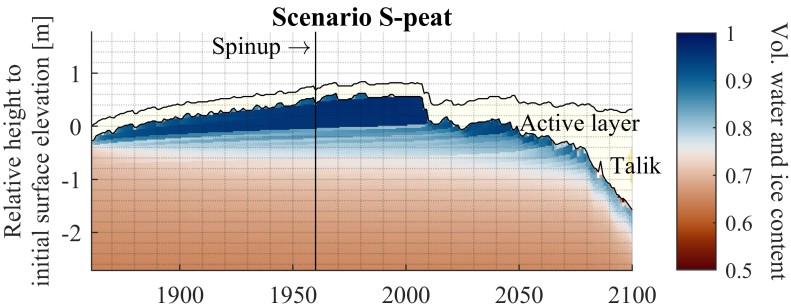

**Figure 11.** Volumetric water and ice content in the permafrost on the reference date August 31 in model scenario *S-peat*. Dark blue colors indicate an increased volumetric water and ice content at the top of the permafrost, where ice segregation takes place. Values in the thawed layer are not displayed. The spin-up covers the years 1860 to 1960. The sum of volumetric water and ice content is shown as soil water can still occur below freezing temperatures dependent on the soil type. For scenario setup see Table 4. The results show a significant formation of segregated ice below the permafrost table and associated ground heave.

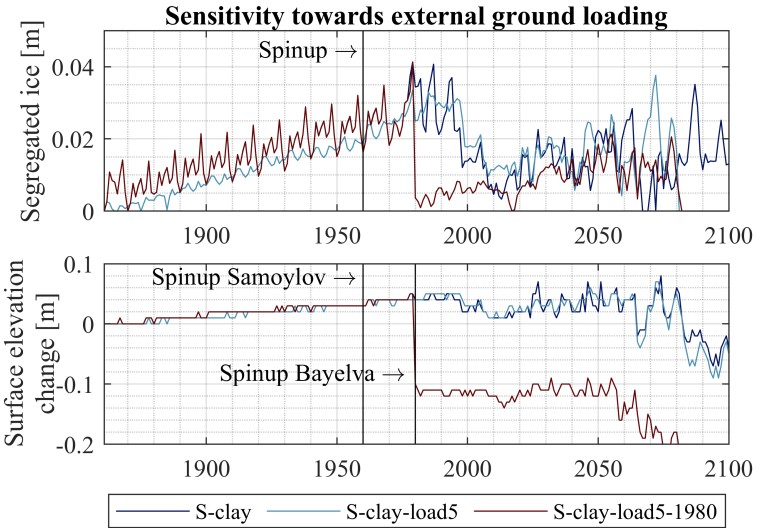

**Figure 12.** Column-accumulated segregated ice and surface elevation changes on the reference date August 31 for the model scenarios *S-clay*, *S-clay-load5* and *S-clay-load5-1980*. A load that is acting on the soil column from the beginning of the simulation (*S-clay-load5*) suppresses ice segregation. A load that is added during thaw consolidation (*S-clay-load5-1980*) accelerates the process and the ground surface subsides faster.

## 3.6 Sensitivity towards sedimentation

We analyze the effect of sedimentation on ice segregation in the ground. For this purpose, we compare scenarios with a constant sedimentation rate of $2 \, \mathrm{mm \, a^{-1}}$ applied for different durations of model spin-up. As sedimentation is only applied when the near-surface layer is unfrozen (Sect. 2.1.4), the effective sedimentation rate is about $0.55 \, \mathrm{mm \, a^{-1}}$. The model scenario without sedimentation (*S-clay*) and with sedimentation over a spin-up of 100 years (*S-clay-sed100*) show only small differences in ice segregation with $0.038 \, \mathrm{m}$ compared to $0.034 \, \mathrm{m}$ in scenario *S-clay*. Significantly thicker ice-rich layers form when simulating longer periods with ongoing sedimentation and extending the spin-up to 500 years and 1000 years, respectively (*S-clay-sed500*, *S-clay-sed1000*): Figure 13 shows that the segregated ice increases linearly with the time that the sedimentation is applied. The permafrost table gradually rises upwards with rising surface elevation due to sedimentation and new segregated ice can continuously form. In the 1980s, the segregated ice (elevation change of the ground surface) reaches 0.038 (0.085) m for *S-clay-sed100*, 0.122 (0.307) m for *S-clay-sed500* and 0.310 (0.582) m for *S-clay-sed1000*. The elevation change of the ground surface can be attributed to the formation of segregated ice and the deposited material. Soil swelling in the active layer accounts for only a small percentage of the change.

Due to the formation of segregated ice during the cold climate conditions of the model spin-up, sedimentation also influences the thaw trajectories under a warming climate. As more segregated ice has to be melted, the process takes longer time in the simulations with sedimentation. While the main layer of segregated ice is melted already in the 1990s for *S-clay* and *S-clay-sed100*, it disappears for *S-clay-sed500* around 2010 and for *S-clay-sed1000* around 2020. The development of a talik is not affected by the sedimentation as it occurs the first time in the 2080s. Yet, while *S-clay* and *S-clay-sed100* subside by 0.082 m and 0.056 m until the end of the 21st century, *S-clay-sed500* and *S-clay-sed1000* show more subsidence with 0.188 m and 0.286 m, although sedimentation continues throughout the entire simulation. The higher values can be mainly related to the larger amount of segregated ice that is melted but also to the stronger compaction of the soil layer due to higher total stress caused by the deposited material.

A similar behaviour is found for the Bayelva field site, represented by the model scenario *B-clay-sed1000*. It builds up 0.267 m of segregated ice, which is slightly lower than the simulated values at Samoylov island with 0.310 m in model scenario *S-clay-sed1000*. Also here, the melting of the ice-rich layer is delayed and disappears around 2020, similar to *S-clay-sed1000*.

We also investigate the effect of the sedimentation sedimentation rates on the accumulation of segregated ice. For the reference runs *S-clay-sed500* and *S-clay-sed1000* (Fig. 13), the ice-enriched layer at the at the top of the permafrost contains a volumetric water and ice content of 67 to 69 % and a volumetric segregated ice content of 12 to 14 %. These values are significantly reduced to 60 % and 5 % when increasing the sedimentation rate by a factor of three as performed in model scenario *S-clay-sed350-3x* (Fig. S5 in Suppl. S3). This is likely due to the faster rising permafrost table, which does not allow for more ice accumulation at the top of the permafrost. The same trend is detected when doubling the sedimentation rate at the Bayelva field site: While *B-clay-sed1000* simulates a volumetric water and ice content in the ice-rich layer of 73 % and a volumetric segregated ice content of 18 %, the same layer reaches only 69 % and 14 %, when the sedimentation is doubled in model scenario *B-clay-sed370-2x*.

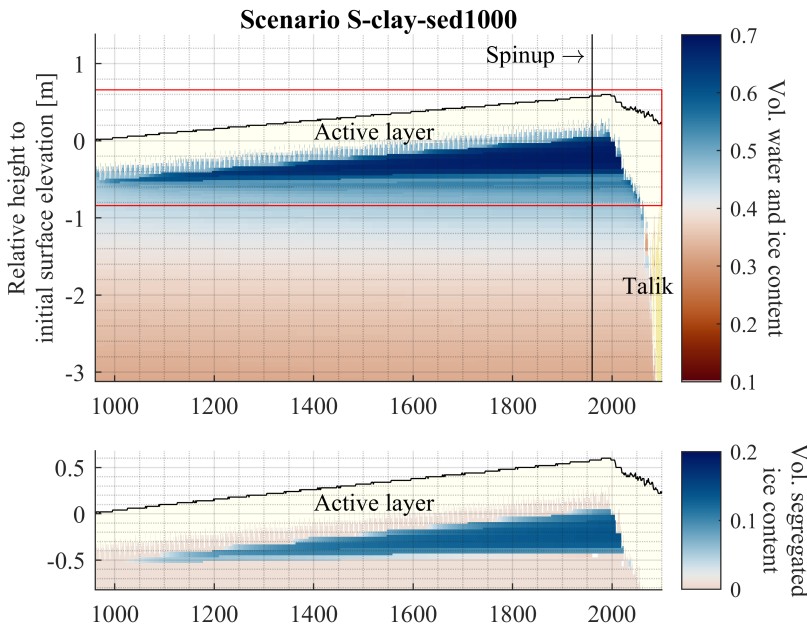

**Figure 13.** Volumetric water and ice content and volumetric segregated ice content on the reference date August 31 for the model run *S-clay-sed1000*. Dark blue colors indicate an increased volumetric water and ice content at the top of the permafrost, where ice segregation takes place. Values in the thawed layer are not displayed. The sum of volumetric water and ice content is shown as soil water can still occur below freezing temperatures dependent on the soil type. For scenario setup see Table 4. The results show a significant formation of segregated ice below the permafrost table under a constant sedimentation rate. The ground surface rises due to both accumulated material and the build-up of segregated ice. With the deepening of the active layer, the segregated ice is melted and thaw consolidation takes place with associated ground subsidence.

# 4 Discussion

## 4.1 Ice segregation and thaw consolidation in CryoGrid

Previous work with CryoGrid (Westermann et al., 2016, 2023) did not consider soil mechanical processes when modelling the thermal regime of permafrost environments. This corresponds to the assumption of a time-constant soil porosity, so that e.g. the effect of ground loading cannot be represented. However, CryoGrid already provides a stratigraphy class to include excess ice (Westermann et al., 2023). Earlier studies focussed on simulating ground subsidence upon thawing with CryoGrid 3 in the Samoylov area in a 1D setting (Westermann et al., 2016), and the thaw-induced transition from low- to high-center polygons 565 with laterally coupled tiles (Aas et al., 2019; Nitzbon et al., 2019). Here, a fixed amount and stratigraphy of excess ice is defined at the beginning of the simulation, which is not necessarily in equilibrium with climatic conditions. Upon thawing, excess water is mobilized and routed away without considering changing hydraulic properties of the ground due to the fixed porosity.

Observation sites in permafrost environments are scarce and ground ice contents are poorly constrained (Cai et al., 2020). Therefore, model schemes like those of Nitzbon et al. (2019, 2020) and Westermann et al. (2016) are associated with significant uncertainties when applied at sites without data on ground ice content. Furthermore, thermokarst simulations conducted with prescribed excess ice contents are highly sensitive to the initial depth of the excess ice layer, and small changes to the model settings strongly affect the onset of excess ice melt and thermokarst development.

In contrast, the new model scheme allows us to evaluate the full "ground ice mass balance" including both formation and melt, depending on defined soil mechanical properties and the applied climate forcing. The porosity is adjusted in each time step according to stress conditions in the ground and the soil characteristics (initial void ratio, residual stress, compression index). Therefore, it evolves in time in response to the model forcing, which can lead to for example saturated ground conditions and thus soil swelling. Furthermore, the ice content decreases with increasing confining pressure due to stronger compaction. As the mod scheme can form segregated ice according to the climatic forcing and soil properties, the segregated ice content is not a model input anymore, but is calculated by the model itself during spin-up. While we have only used a ten-year period in the mid 20th century for model spin-up in this study, the future goal is to develop improved spin-up procedures with which it is possible to simulate realistic cryostratigraphies, in agreement with observations (see Sects. 4.4 - 4.6).

## 4.2 Influencing factors for ice segregation and thaw consolidation

The presented model scheme is capable of simulating ice segregation and thaw consolidation, which are both governed by a strong seasonality. Segregated ice forms during fall refreezing at both the downwards penetrating freeze front near the surface, and the upwards going freeze front at the bottom of the active layer. This leads to ice-enriched zones both near the surface and at the bottom of the active layer. When the ground thaws in spring and summer, the segregated ice in the upper parts of the active layer melts rapidly. In contrast, the ice at the bottom of the active layer only melts at the end of the thaw season or might even persist and eventually become part of the permafrost. Over time, this leads to an ice-enriched zone below the permafrost table (Fig. 4). With the deepening of the active layer, the segregated ice begins to melt and excess pore water pressures initiate the consolidation process. As this process takes a certain amount of time, soil stability can potentially be reduced and excess pore water pressures might cause active layer detachment slides in sloping terrain (Lewkowicz and Harris, 2005). During thaw consolidation, the void ratio is reduced according to the compression curve of the soil and the ground surface subsides.

We highlight that the new model scheme allows us to analyze the ground ice content both in space and time and ground properties such as porosity and ice content are adjusted in each time step according to the stress conditions in the soil column. We identify several factors, which influence the soil mechanical processes: (i) soil water contents and ground temperature gradients, (ii) soil type and (iii) external ground loading, which will be discussed in the following.

(i) The climatic forcing controls ice segregation with its influence on ground temperatures and therefore the active layer thickness as well as the water supply by rain- and snowfall and evaporation rates. Moist conditions lead to more segregated ice compared to dry conditions (Fig. S1 in Suppl. S2), as more water is available for ice segregation and the increased hydraulic conductivity supports the mobilization of soil water. Furthermore, high temperature gradients in the soil column (Fig. S3 in Suppl. S2) enhance ice segregation due to the temperature dependency of the soil matric potential in frozen state (Westermann

et al., 2023). This likely increases the formation of segregated ice in continental settings with large differences in ground temperatures between summer and winter season.

(ii) The soil properties have a strong influence on ice segregation. One reason is different hydraulic properties, which govern the matric potential and its dependency on water and ice contents (Westermann et al., 2023). In addition, the soil types feature different freezing characteristics ($\alpha$ and $n$) controlling the unfrozen water content for negative ground temperatures. For sand, the matric potential only becomes strongly negative, when almost all water is frozen. In this state, however, the hydraulic conductivity is already very low, so that little or no water is drawn to the freezing grid cell. As a consequence, the formation of segregated ice is negligible. For the soil types silt, clay and peat, the matric potential already becomes strongly negative, when there is still a significant amount of unfrozen water available. The hydraulic conductivity is hence high enough so that water is drawn to the freeze front, resulting in ice segregation. When segregated ice is melted, the behaviour of the soil is controlled by the compression curve, which is defined by the parameters initial void ratio, residual stress and compression index. The model shows that the reaction of a soil to changing climatic conditions is strongly affected by its properties.

(iii) External loads suppress the formation of segregated ice depending on the weight of the external load. This can be explained by an increased total stress in the soil column and consequently an increased effective stress on the soil skeleton, which needs to be overcome by the matric potential differences in order for segregated ice to form. These stress conditions also speed up the consolidation process as the excess water from the melting of segregated and pore ice ice is pressed out of the soil matrix faster due to the higher total stress. This is especially true for the first year after the external load is applied: The load is acting on the top of the permafrost, where the segregated ice accumulated in the past years. Due to the selected soil characteristics of the clay in our model setup (Table 3), mobile soil water occurs next to the ground ice and is pressed out due to higher total stress. In addition, the reduced porosity due to soil consolidation cause an increase in thermal conductivity, allowing the active layer to thaw to greater depths. Therefore, additional ground ice near the top of the permafrost may melt as well.

In conclusion, the model results highlight that ice segregation and thaw consolidation are influenced by climatic factors, site-specific factors (soil properties) and anthropogenic factors such as deployment of structures causing external loading.

### 4.3 The impact of sedimentation on ice segregation

Including sedimentation in the model runs allows us to analyze the aggradation of segregated ice in syngenetic permafrost, which in particular for fine-grained soil material contains ice-rich layers and is susceptible to ground subsidence upon thawing. When material is deposited on the ground, either by sedimentation (e.g. in river deltas or at the bottom of slopes) or as organic matter through vegetation growth, the permafrost table moves upwards under consistent climatic conditions. The shift of the freezing front leads to a thickening of the ice-rich soil layers as demonstrated in this study (Sect. 3.6). Our results show that longer periods of sedimentation result in thicker segregated ice layers, with the sedimentation rate influencing the overall ice content of the accumulated layers. The segregated ice content decreased with increasing sedimentation rates in our simulations, likely since the rise of the permafrost table due to sedimentation "outpaces" the accumulation of segregated ice at the top of the permafrost.

The amount and distribution of ground ice are strong controls on the thaw behaviour under a warming climate. A thin layer of segregated ice below the active layer melts rapidly and causes ground subsidence and possibly positive feedbacks to thawing, even for moderate warming. However, ground ice in deeper soil layers needs more intense warming or longer time periods until thawing and its consequences occur. Consequently, it is not only important how much excess ice can be found in the ground, but as well how it is distributed in the soil column. A setup in our new model scheme driven by paleo-climate data and sedimentation rates may be able to form segregated ice in different depths, improving estimates of ground ice content especially in field sites with poor field observations of ground ice stratigraphies (see Sect. 4.4).

## 4.4  Comparison to in situ observations of ground ice

The primary goal of this study is to demonstrate the accumulation and melt of segregated ice in CryoGrid and explore its sensitivity to different model settings. However, it is meaningful to compare the simulated ground ice contents to in situ observations, at least for some of the model scenarios and in a semi-quantitative way, keeping the limitations of the model setup in mind. For most model scenarios without sedimentation, only thin layers of ground ice are formed during the model spin-up, making a comparison to situ observations, which generally report ice contents for thicker layers, challenging. However, we can use the sedimentation runs for a comparison with observed ground ice contents, in case the sedimentation history and rates can be sufficiently constrained.

At the Bayelva site, concurrent observations of sedimentation history and ground ice are not available, but we can compare the model results to observations from Adventdalen, Svalbard, located about $120\,\mathrm{km}$ to the south-east, which features broadly similar climate characteristics. Cable et al. (2018) present cryostratigraphic observations including excess ice contents from sediment cores, and we select a core (A2a) featuring Holocene deposits with high contents of fine-grained material and only low organic content, which is similar to the simplified stratigraphy in the *B-clay* scenarios. The deposits have been formed by aeolian sedimentation with sedimentation rates of $1.1\,\mathrm{mm\,a^{-1}}$ in the last 600 years (Cable et al., 2018), which is the same as the effective sedimentation rate in the *B-clay-sed370-2x* scenario. The observed volumetric moisture content in the aeolian deposits of the selected sediment core is approximately 76 % (calculated from the gravimetric moisture content of 54 %), with a volumetric excess ice contents of 18 %. These values are in a similar range as the simulated ice contents in the ice-enriched layer in *B-clay-sed370-2x*, with total volumetric moisture contents of up to 69 % and volumetric segregated ice contents of up to 14 %.

For the Samoylov site, a comparison to ground ice observations is challenging, as both wedge ice and segregated ice have contributed to the build-up of syngenetic permafrost during the Holocene (Schwamborn et al., 2002). Furthermore, the island has a complex sedimentation history, being located in the upper part of the vast delta of the Lena river. Radiocarbon dates of organic material in a core from Samoylov island are presented in Schwamborn et al. (2002), suggesting that approximately $5\,\mathrm{m}$ near the top of today's permafrost were deposited during 2000 to 2500 years. Another core is described in Schwamborn et al. (2023), showing calibrated radiocarbon ages of 1370 ky BP near the top of the permafrost (0.5 to 0.6 m depth below the surface), while the ages varied randomly between 4000 and 6000 ky BP between 1.5 and 6.5 m depth. While this shows that sedimentation has not been a continuous process on Samoylov island, with sediments likely being reworked, we can

broadly estimate effective sedimentation rates between $0.3 \ \mathrm{mm \, a^{-1}}$ and $2.5 \ \mathrm{mm \, a^{-1}}$ for the permafrost section underlying the present-day active layer. For the Samoylov site, long-term sedimentation runs have been performed for effective rates of $0.55 \ \mathrm{mm \, a^{-1}}$ (*S-clay-sed1000*) and $1.7 \ \mathrm{mm \, a^{-1}}$ (*S-clay-sed350-3x*), which is well in the range that can be established from the drill cores. We compare the simulated ice contents to the cryostratigraphy of a polygon center where segregated ice indeed occurs, synthesized from observations on Samoylov island for modeling purposes (Nitzbon et al., 2019). For the ice-rich layer

in the top-most permafrost (below 0.9 m depth in Nitzbon et al., 2019), a total volumetric ice content of 65 % is prescribed, with a volumetric excess ice content of 18 %. In comparison, *S-clay-sed1000* features a total volumetric ice content of 69 % (volumetric excess ice content 14%), and *S-clay-sed350-3x* shows a volumetric ice content of 60 % (volumetric excess ice content 5%). Overall, it is encouraging that the simulations produce ice contents in the correct order of magnitude also for Samoylov island. However, we emphasize that this comparison is highly uncertain and should not be understood as a validation

of the model, in particular since formation of segregated ice occurs in concert with wedge ice on Samoylov island.

    Nevertheless, the comparison to observed ground ice contents suggests that our model (with the simplified stratigraphy) is capable of modelling ice segregation in the correct order of magnitude at both the Bayelva field site and Samoylov island. However, we see three main limitations of our model setup which need to be addressed to simulate more realistic cryostratigraphies: (i) the climate data used for spin-up, (ii) the constant sedimentation regime and (iii) the hydrological regime. Overcoming these

challenges would allow us to simulate the ground ice evolution since the formation of permafrost, including ice segregation at the permafrost base. This would enable to resolve the ground ice distribution also in greater depths in the soil column, e.g. the ice accumulations in glacial and lacustrine sediments (Gaanderse et al., 2018; Smith et al., 2007; Wolfe and Morse, 2017). (i) Simulating ground ice contents at a specific field site requires long-term historical forcing data and our spin-up procedure with a repetition of a 10 years period in the 20th century is not suitable to represent the historical climate. This limitation can

in principle be resolved by preparing time series of model forcing from paleo-climate model runs. (ii) We applied a constant sedimentation rate, while the sedimentation regime experiences changes over such long time periods, which are not represented in the current model setup. In practice, this problem is hard to resolve, but it may be enough to prescribe sedimentation rates for the most important layers within a cryostratigraphy. Moreover, "sedimentation" with organic material could be simulated by a dedicated subsurface carbon cycle model. (iii) The hydrological regime of polygonal ground is different to the 1D setup

in our study. Previous modeling studies have shown that the seasonal dynamics of ground temperatures and water/ice contents in the polygonal tundra landscape cannot be captured with one-dimensional simulations (Nitzbon et al., 2019), as conducted in this study, which likely affects the accumulation of segregated ice. Our model setup could in principle be included in three-dimensional simulations with laterally coupled tiles (Aas et al., 2019; Nitzbon et al., 2019) without taking lateral cryosuction into account (Sect. 4.5). However, realistic long-term simulations would require the additional representation of wedge ice

build-up, which is the primary process of ground ice accumulation in polygonal tundra landscapes.

    Another possibility for validation could be laboratory freezing experiments. An example is the study by Xue et al. (2021), who conducted a one-sided freezing experiment in saturated soil to investigate the relationship of matric potential, unfrozen water content and segregated ice. Further experiments have been presented by Konrad and Morgenstern (1980, 1981, 1982a, b); Smith and Onysko (1990); Williams and Wood (1985). The experimental conditions could be applied to a model setup and

results in matric potential and in water/ice content could be compared. However, several soil parameters, which are needed as model input, are often not given in the experiment, such as initial void ratio, residual stress and compression index, so that a meaningful comparison might be challenging. Nevertheless, model validation with laboratory experiments could be conducted in future studies and will likely contribute to model improvements.

## 4.5 Limitations and uncertainties

In this study, we test the sensitivity of a new model scheme for ice segregation and thaw consolidation towards a range of climatic and environmental factors. However, there are limitations and uncertainties which are important to address in future studies.

In its present form, the model is one-dimensional and hence does not account for lateral processes, e.g. lateral water flow due to cryosuction. This approach is feasible for our proof of concept or when simulating field sites with little lateral variability in the soil structure. In order to consider complex lateral processes, laterally coupled tiles as applied in Martin et al. (2021); Nitzbon et al. (2019, 2020) are necessary.

The buoyancy effect reduces the stress on the soil matrix in case the soil is at or close to saturation, while the soil skeleton carries the weight of the overlying soil layers under dry conditions. For a continuous transition, we scale the buoyancy effect between 50 % and 100 % saturation (Sect. 2.1.1). The threshold of 50% is based on an ad-hoc assumption, which should be investigated in more detail in future studies.

Our model does not take into account the different densities of water and ice. In dry conditions, this effect does not influence the void ratio of the soil significantly as the ice can expand into free pore space. However, in saturated conditions, the volume change of the phase change of water can change the void ratio of the soil. The associated error accounts for approximately 8 %, so that a result of 0.1 m segregated ice formation would correspond to about 0.11 m including the effect of phase change. Consequently, the volumetric ice content and the corresponding segregated ice and associated surface heave is slightly underestimated in our model. At least phenomenologically, this could be implemented as an independent process, which does not interact with the calculation of soil mechanics and water fluxes. However, as CryoGrid assumes the density of ice to be equal to the density of water in a range of submodules, e.g. for the snow cover (Westermann et al., 2023), this has not been implemented yet for consistency reasons.

Creep processes can play an important role in permafrost and can occur at very low slope angles (Williams and Smith, 1989). The soil mechanical processes implemented in the model consider only primary consolidation, and do not account for long-term creep processes, which can be considered to be a first order approximation. For long-term simulations with thick sedimentary deposits near or above thawing temperatures, creep processes should be implemented to get a better representation of the deformation of the soil column.

Furthermore, we simplify the relationship between void ratio and effective stress by having a constant compression index for each soil type. As a consequence, the deformation of the soil is completely reversible. This implies that a soil, which is compressed e.g. by external loading, expands to its initial state upon unloading. This simplifying assumption also affects the quantification of segregated ice in the model, which is defined as the ground ice volume that exceeds the initial pore volume

(initial void ratio) of the soil. An improved representation of the soil system and formation of segregation ice could therefore be obtained by introducing the concept of a preconsolidation stress, and a recompression index for unloading and reloading branches of the void ratio vs. effective stress curve. For the model cases presented in this paper, these simplifications are expected to have a minor impact on the modelled ground surface deformation, as processes such as sedimentation and build-up of ground ice result in an increase in effective stress and thus a movement in the normally consolidated domain. Unloading effects are minor in the presented model cases and result primarily from changes to the water balance and drainage of excess water due to ground ice melt in the last decades of the model scenarios. In potential modelling scenarios including erosional processes, the proper handling of unloading and reloading would be essential.

In this study, we run the model for syngenetic permafrost, i.e. assuming frozen conditions at the beginning of the simulation and simulating the evolution under a sedimentation regime. Like this, the ground ice distribution in the uppermost parts can be obtained. The ground ice in deeper soil layer could be modelled in principle with an extended spin-up period. In contrast to that, epigenetic permafrost could be simulated through a model setup with unfrozen conditions at the beginning. In this case, ground ice would likely be formed at the top of the permafrost and at the base of the advancing freeze front, but three-dimensional effects such as lateral water fluxes and non-horizontal freeze fronts might play a major role and are not accounted for in the current model. Furthermore, both for syngenetic and epigenetic permafrost, plaeo-climate data to force the model (and potentially information on the sedimentation regime) are required to obtain realistic ground ice distributions (Sect. 4.4).

## 4.6 Possible applications

The presented study is a proof of concept of a new model scheme for the CryoGrid community model. Applying it to different study setups could contribute to the demonstration of natural processes in permafrost soils.

Soils with a high organic content are widespread in the Northern Hemisphere (Hugelius et al., 2014). As these soils are sensitive to changing stress conditions due to a high compression index, it could be beneficial to apply the new model scheme for field sites with high carbon contents in the soil stratigraphy. The accumulation of peat could be simulated with the associated ice segregation in long-term runs. One model target could be the evolution of peat plateaus. When peat mires are exposed to cooling climate conditions e.g. during the Little Ice Age (Kjellman et al., 2018), epigenetic permafrost develops. Due to the soil properties of peat, we find in this study that considerable amounts of segregated ice can be formed (Sect. 3.4), resulting in strong ground subsidence under a warming climate, in agreement with observations from peat plateau sites (e.g. Martin et al., 2021).

In addition to peat plateaus, the new model scheme could be a step forward to simulate polygonal tundra. To do so, a model for wedge ice formation is required. It could be coupled laterally to the polygon center (see Martin et al., 2021; Nitzbon et al., 2019, 2020) represented by the stratigraphy class in this study, which could accomplish the segregated ice formation occurring potentially in the polygon centers.

Furthermore, applying the model to inclined terrain could enable the user to simulate conditions on slopes affected by permafrost. If segregated ice is melted rapidly and excess water is mobilized faster than it can be drained, high pore pressures can develop. This can promote weak layers that can shear off in form of active layer detachment slides (Lewkowicz and

Harris, 2005). Analyzing the distribution of stress conditions would be possible with the new model scheme, enabling to detect potential weak layers in the slope. Furthermore, water fluxes along the slope could be initiated by comparing the hydraulic head of the soil columns.

The new model scheme could as well be applied in geotechnical studies, especially as the effect of external loads on the soil parameters such as porosity can be analyzed. Climate change can warm permafrost temperatures and increase active layer thickness or even lead to the development of a talik. The resulting subsidence rates could be simulated with the model approach and contribute to the analysis of the stability of settlements and infrastructure.

## 5 Conclusions

In this study, we present a new model scheme for the CryoGrid community model, which demonstrates ice segregation as well as thaw consolidation. The model is capable of building up layers of segregated ice with associated ground heave and calculating subsidence upon thawing, which can improve simulations of ice-rich ground responding to a warming climate. It combines soil mechanical processes with soil hydrology according to Richards equation and soil freezing characteristics. We run the model with forcing data of two permafrost sites (Samoylov, Siberia and Bayelva, Svalbard) and analyze different influencing factors such as climatic forcing, soil type, external loads and sedimentation. Our main conclusions are:

– The new model scheme is able to calculate climate-driven ice segregation and thaw consolidation in permafrost environments. By doing so, not only the amount of ground ice can be simulated but also its distribution in the soil column. These processes lead to ground surface heaving and subsidence, respectively.

– The model shows that important controlling factors on ice segregation and thaw consolidation can be simulated such as (i) ground temperature gradients and soil water content, (ii) soil type and (iii) external loads. (i) Large gradients in ground temperatures, as well as high soil water contents in the active layer through high precipitation support ice segregation. (ii) Fine-grained soils lead to an increased amount of segregated ice compared to coarse sediments such as sand, where insignificant reactions are detected. Peat shows a more significant formation in segregated ice, emphasizing the sensitive behaviour of organic-rich soil. (iii) External loads can suppress ice segregation and lead to a speed up in thaw consolidation.

– Taking into account sedimentation, the permafrost table and thus the thaw front shifts upwards with the ground surface. This way, thick layers of segregated ice can be generated by the presented model scheme, which lead to strong subsidence when climate conditions become warmer.

While the new model scheme cannot yet account for other pathways of excess ice formation (such as wedge ice), the implementation of segregated ice represents a first step towards the understanding of the formation of ground ice and thaw consolidation under a warming climate and can help to determine the distribution of ground ice in depth and time.

*Code availability.* The model code is archived on Zenodo (https://doi.org/10.5281/zenodo.6884775, Aga, 2022).

*Author contributions.* JA planned the concept of the study, developed the model code, analyzed the simulations and wrote the manuscript
including all tables and figures. SW and TIN contributed with ideas as well as technical and organizational support. SW and ML designed the
overall concept and structure of the CryoGrid community model. ML provided forcing data for the future simulations. JB provided climate
data of the Samoylov station and Bayelva station as well as measurement data for validation. All authors contributed to the final manuscript
with input and suggestions.

*Competing interests.* The authors declare that they have no conflict of interest.

*Acknowledgements.* We acknowledge the funding by EU Horizon 2020 (Nunataryuk, grant no. 773421), the Research Council of Norway
(PCCH-Arctic, grant no. 320769) and ESA (Permafrost_CCI, https://climate.esa.int/en/projects/permafrost/).

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
