# Peer review of "Simulating ice segregation and thaw consolidation in permafrost environments with the CryoGrid community model"

_EGUsphere, 2023_

## Author Comment (AC1)

**Response to anonymous referee 1**

We would like to thank the reviewer for the detailed and useful comments, which have helped to improve the quality and readability of our manuscript. In the following, we provide a reply to the points discussed by the reviewer as well as changes in the manuscript.

5 The comments of the reviewer are written in **bold**, the extracts of the manuscript in *italics* with changes highlighted in blue and line numbers referring to the revised manuscript.

**This paper describes a new model scheme, in a suit of the CryoGrid community model (version 1.0. Under review at GMD), to simulate the temporal evolution and the vertical distributions of ground ice content by calculating ice**
10 **segregation (excess ice) when cold and thaw consolidation when warm, and associated ground heave and subsidence. The model incorporates soil mechanical processes, soil hydrology, and soil freeze/thaw physics. The authors conducted a series of proof-of-concept examinations of the new scheme with respect to climatic (i.e., thermal, and hydrological) conditions, soil types, external loadings, and sedimentation, which demonstrated reasonable performance of the model. It is not so simple to evaluate an additional module when the base model is still under review, however, the reviewer**
15 **found that the manuscript is moderately well organized and written. Yet, some elaborations and clarifications regarding the points raised below will improve the manuscript before being published in the Cryosphere journal.**

The base model is now published and we replaced the reference of the preprint with the published version:

20 Westermann, S., Ingeman-Nielsen, T., Scheer, J., Aalstad, K., Aga, J., Chaudhary, N., Etzelmüller, B., Filhol, S., Kääb, A., Renette, C., et al.: The CryoGrid community model (version 1.0)–a multi-physics toolbox for climate-driven simulations in the terrestrial cryosphere, Geoscientific Model Development, 16, 2607–2647, https://doi.org/10.5194/gmd-16-2607-2023, 2023.

**Ll. 144-146: When it is referred by a general term "soil", is it assumed that the saturation (in terms of volumetric**
25 **water content?) is equal among the constituents (e.g., mineral, organic)?**

The saturation is calculated separately for each grid cell of the soil column and the soil water is distributed equally within the grid cell. In the CryoGrid community model, each grid cell can only consist of one soil type (sand, silt, clay or organic), which cannot be mixed within the gridcell. Consequently, changes in the soil stratigraphy have to be reflected in the grid cell
30 sizes.

Line 165: *To compute the pore water pressure, the buoyancy effect has to be accounted for. For each grid cell of the soil column, which is composed of one soil type (sand, silt, clay or organic), the saturation is calculated separately. When the soil*

*is at or near saturation, the porewater results in a reduction of the stress on the soil matrix, as the density of each component*
35 *is reduced by the density of water.*

**L.146: How do you justify the threshold value of 50%? Some reference or practice examples would be helpful.**

The threshold of 50 % is an ad hoc assumption for a pragmatic implementation of the buoyancy effect in the code. This is of
40 course a source for uncertainty, so that we included a paragraph in chapter 4.4.

Line 708: *The buoyancy effect reduces the stress on the soil matrix in case the soil is at or close to saturation, while the soil skeleton carries the weight of the overlying soil layers under dry conditions. For a continuous transition, we scale the buoyancy effect between 50 % and 100 % saturation (Sect. 2.1.1). The threshold of 50% is based on an ad-hoc assumption,*
45 *which should be investigated in more detail in future studies.*

**L. 154: Does "grain" refer to mineral only, or include organic matters in the module (or CryoGrid) terminology?**

The term "grain" refers to the mineral and organic matter in this context. To avoid confusion, we changed the wording in the
50 manuscript.

Line 176: *The compressibility of a soil can be expressed with a linear relationship between void ratio $e$ [-] (defined as the ratio of pore volume to the volume of mineral and organic matter) and the decadic logarithm of the effective stress $\sigma'_z$ [Pa] (Murthy, 2022) as illustrated in Fig. 3.*
55

**L. 174, l. 211, and l. 454: "section 2.1" is used for a reference, however, section 2.1 includes many subsections. It is more reader-friendly to provide more specific reference (such as at l. 241 "section 2.1.2").**

Line 174: As we are referring to section 2.1, not any of the subsections, it is not possible to refer to a specific subsection.
60
Line 211: We realized that the reference is unnecessary in this place an deleted it.

Line 454: We changed the reference to the subsection.

65 Line 514: *As the active layer is saturated with 70-80 %, the compression takes places immediately as described in Sect. 2.1.1 for unsaturated conditions.*

**Ll. 205-207: It is not clear how pore water pressure is updated with respect to eqs. (4) and (9).**

We improved the clarity in the revised manuscript.

In our model, we calculate excess pore water pressures, while pore water pressures under normal conditions are estimated with the buoyancy effect as described in line 165 ff. As we know the total stress of the overlying soil, we can solve equation 4 for the effective stress of the soil. The value of the effective stress then determines the void ratio (through the compression curve, equation 6), so that the thickness of the grid cell can be calculated.

If the total stress is increased, e.g. by applying an external load, the additional total stress will be taken up completely by the soil water in the first time step. As the effective stress stays unchanged in this time step, we can solve equation 9 for the excess pore water pressure $u_e$ as we know the value for the total stress and the (unchanged) effective stress. The development of excess pore water pressure will result in water fluxes out of the grid cell (equation 10).

When water is flowing out of the affected grid cell, the grid cell looses volume and therefore the thickness d is reduced as described in equation 11. As grid cell thickness d is now known, we can solve equation 8 for the porosity, equation 7 for the void ratio and consequently equation 6 for the effective stress. The effective stress increases and the excess pore water pressure reduces until it reaches a value of zero. In summary, under saturated conditions, we can invert equation 6-8 and update the excess pore water pressure like that.

Line 227: *If the soil is unfrozen and saturated, [...]* In these conditions, the effective stress $\sigma'_z$ [Pa] on the soil skeleton can be calculated by reducing the total stress $\sigma_z$ [Pa] by the pore water pressure u [Pa], as shown in Eq. 4.

*When external loads are applied or sedimentation takes place, the additional stress is taken up by the soil water in a first step, leading to excess pore water pressures $u_e$ [Pa]:*

$$u_e = \sigma_z - \sigma'_z, \tag{1}$$

*which results in water fluxes $j^v_{w,u_e}$ [$\mathrm{m\,s^{-1}}$] away from the affected grid cell*

$$j^v_{w,u_e} = -\frac{K_w}{\rho_w g}\left(\frac{\partial u_e}{\partial z}\right). \tag{2}$$

leading to a reduction of the excess pore water pressure by consolidation. *The water flux is added to the fluxes calculated based on the Richards equation (Eq. 1). The consolidation continues until the excess pore water pressure reaches a value of zero, while at the same time the effective stress is increased [...]*

*After calculating the water fluxes in the soil column, the change in water content for each grid cell can be derived. For unsaturated conditions, changing water content is replaced by air. For saturated conditions, no air inflow is possible and changes in water content affect the grid cell size. The thickness of a grid cell d [m] can be calculated for saturated conditions from the volume $\Theta$ [$\mathrm{m^3}$] of the grid cell (being the volume of water, ice, mineral, organic) divided by the area A [$\mathrm{m^2}$]:*

$$d = (\Theta_w + \Theta_i + \Theta_m + \Theta_o)/A \tag{3}$$

*With the thickness of the grid cell d being calculated for saturated conditions directly from the change in water content, we can invert Eq. 8 and Eq. 7 and solve Eq. 6 for the effective stress $\sigma'_z$ to update the excess pore water pressure with Eq. 9. A reduction/increase in water content results in a change in effective stress and thus in compression/swelling of the soil.*

**Figure 3: What does the solid straight blue line in the figure (with a filled reverse triangle on the left shoulder) denote? If it explains something it should be noted either in caption or text. Otherwise, should be removed.**

The blue line indicates the ground water level. We added some explanation in the figure to clarify.

[Figure]

*Illustration of the potentials resulting in water fluxes as calculated in the model (not to scale): During winter season, the entire soil column is frozen and no substantial water fluxes occur. When the upper soil layers are unfrozen during summer season, water fluxes in the unsaturated zone are controlled by the gravitational potential $P_g$ and the matric potential $\psi$. Rainwater or meltwater infiltrates from the top to deeper layers. Most important for ice segregation is the matric potential during fall refreezing (marked in red). In case the downward thawing from the surface during summer reaches (partly) the segregated ice, excess pore water pressures $u_e$ develops, leading to thaw consolidation.*

**Ll. 257-258, 278: Description of the range of temperature variations and the period of measurement are somewhat awkward. It is assumed to be meant something like "The observed ground temperature at a depth of 20.75 m varied between -9.0 and -7.9 C during the period of 2007 to 2016."**

We changed the text in the manuscript.

Line 286: *The observed ground temperature at a depth of 20.75 m increased from -9.0 °C in 2007 to -7.9 °C in 2016 (GTN-P, 2018). With these records, the field site shows one of the strongest warming of permafrost within 123 globally distributed boreholes (Biskaborn et al., 2019).*

Line 309: *Observed ground temperatures at a depth of 9 m varied between -3.0 and -2.6 °C during the period of 2007 to 2016 (GTN-P, 2018) and thus show relatively warm permafrost.*

**Ll. 262-263, l. 281: Description is somewhat sloppy. The forcing data should have been taken from "the CCSM4 outputs simulated under the RCP8.5 scenario", or "the RCP8.5 run by CCSM4". Why there is no reference on this model or its CMIP5 outputs?**

We changed the wording according to the suggestion of the reviewer. Furthermore, we added the necessary references.

Line 291: *The climatic forcing between 1960 and 2012 is derived from the reanalysis product CRU-NCEP (combining data from the Climate Research Unit and National Centers for Environmental Prediction, see Kalnay et al., 1996; Harris et al., 2014), downscaled for Samoylov with in situ data from the automatic weather station. After 2012, the forcing data is taken from the Community Climate System Model CCSM4 outputs simulated under the Representative Concentration Pathway (RCP) 8.5 scenario (Meehl et al., 2012).*

**L. 292: It is not clear the "key difference" from what is discussed.**

The key difference in this context is, that we did not assume any excess ice in the beginning of the simulation in contrast to the previous simulation by Nitzbon et al. (2020) and Westermann et al. (2016). We improved the clarity in the manuscript.

Line 324: *For model validation on Samoylov island, we set a soil stratigraphy as described for the center of a low-centred polygon in Holocene deposits in Samoylov in Nitzbon et al. (2020) and Westermann et al. (2016) (multi-layer stratigraphy in Table 3). In contrast to these studies, we did not assume any excess ice in the beginning of the simulation as it is inherently generated by the new model scheme.*

**L. 297: Like the above issue, it is not clear from what the upper 0.15 m was unchanged.**

We clarified this in the manuscript.

Line 324: *For model validation on Samoylov island, we set a soil stratigraphy as described for the center of a low-centred polygon in Holocene deposits in Samoylov in Nitzbon et al. (2020) and Westermann et al. (2016) (multi-layer stratigraphy in*

*Table 3).*

160 Line 329: *To analyze the performance and sensitivity of our model for Bayelva and Samoylov, we used simplified stratigraphies to provide standardized and comparable model setups. Hereby, we use the same soil properties in the upper 0.15 m as in the multi-layer stratigraphy from Samoylov island (see above) to account for the insulating effect of the organic-rich moss layer on top.*

165 **L. 346: it is not so clear what is meant by "the annual downward thawing in summer deepens."**

We meant the increasing active layer thickness. We changed the wording in the manuscript.

Line 386: *Further model scenarios include an external load of 5 kPa during the entire simulation (S-clay-load5) and from*
170 *1980, when the active layer thickness increases (S-clay-load5-1980).*

**L. 361: Is this necessary to mention "that are forced with data from this location"? Any specific reasons?**

We deleted the statement from the manuscript.

175

Line 403: *As the terrain at the field site on Samoylov Island is flat, we set a small gradient of 0.1 m km⁻¹ for the model runs that are forced with data from this location. In contrast, the Bayelva field site is situated on gently sloping terrain, so that we increased the gradient to 1 m km⁻¹.*

180 **Table 3. Why is no value given for maximum snow depth in the B-clay scenario? Does this mean snow depth is unlimited?**

Yes, the snow depth in the B-clay scenario is unlimited. We added this information in the manuscript.

185 Line 400: *For the Bayelva field site, we do not set a limit for the snow depth as the accumulation of the snow resulting from the forcing data generally represents local conditions well (Westermann et al., 2023).*

**Ll. 401-408: It is difficult to read changes in ground surface height from figures 6, and 1-2 in Supplement B so that it is not easy to follow the statement and discussions. Further, for the sake of reader-friendly discussions on evolution and**
190 **relative impacts of ground heave (subsidence) and ice segregation (thaw consolidation), it is suggested to plot segregation ice and surface height changes together (possibly overplotted in figures 7, 8, and 10?).**

We added plots of the surface elevation changes in figures 7, 8 and 10 (old numbering), as suggested by the reviewer. Furthermore, we extended the result sections with the respective information.

195

Line 464: *Figure 9 shows the formation of segregated ice and changes in surface elevation for the model runs S-clay (Samoylov forcing data), S-clay-rain50 (Samoylov forcing data with 50 % rainfall) and B-clay (bayelva forcing data).*

Line 466: *The reduction of rainfall in S-clay-rain50 leads to less segregated ice during the spin-up on average with 0.025 m*
200 *compared to 0.034 in S-clay and consequently less ground heave.*

Line 470: *Furthermore, drier conditions lead to less swelling in the active layer, so that fluctuations in surface elevation, especially during the time period 2020-2100, are dampened (Sect. 3.2).*

205 Line 480: *Despite less ice segregation, the ground heave is more pronounced in Bayelva during the spin-up, due to wetter conditions and a deeper active layer, resulting in stronger soil swelling. During the time period 2020 to 2100, B-clay builds up more segregated ice in deeper soil layers. This can be explained by a slower increase of the active layer thickness compared to Samoylov Island, so that the permafrost table stabilizes for longer time period, enabling new formation of segregated ice. As a consequence, less ground subsidence takes place for the Bayleva run.*

[Figure]

210 *Column-accumulated segregated ice and surface elevation change on the reference date August 31 of each year in the simulation, for the model scenarios S-clay, S-clay-rain50 and B-clay. Drier conditions lead to less formation of segregated ice and thus less ground heave. The moist conditions in B-clay are compensated by smaller temperature gradients in the ground so that similar segregation ice contents are formed during model spin-up.*

215 Line 504: *Soil swelling and shrinking influences the surface elevation, especially during the time period 2020-2100 (Sect. 3.2), but both model scenarios S-clay and S-silt result in a net subsidence of the ground surface.*

[Figure]

**Sensitivity towards soil type**

Column-accumulated segregated ice *and surface elevation changes* on the reference date August 31 for the model scenarios S-sand, S-silt, S-clay and S-peat. With decreasing particle diameter, the soil builds up more segregated ice under equal condi-

220 tions. Peat soils form substantially more ground ice than mineral soils.

[Figure]

*Column-accumulated segregated ice and surface elevation changes on the reference date August 31 for the model scenarios S-clay, S-clay-load5 and S-clay-load5-1980. A load that is acting on the soil column from the beginning of the simulation (S-clay-load5) suppresses ice segregation. A load that is added during thaw consolidation (S-clay-load5-1980) accelerates the process and the ground surface subsides faster.*

**Figures 6, 9, 11 and 1-4 in Supplement B: The vertical axis of altitude seems to be the model surface height (say, absolute altitude). Unless it is necessary (e.g., in laterally coupled cases), it would be easy to follow the height changes if shown by a relative height with reference to the initial surface altitude. Also, the near-surface zones such as active layer is very small, leading it difficult to read and compare especially for the argument of thermal gradients (e.g., ll. 401-408, ll. 420-425). Elaborations will be very helpful.**

We follow the suggestion of the reviewer and used the relative height to the initial surface elevation in the figures 6, 9 and 11 (number 8, 11 and 13 in the revised version). An example is given below. Furthermore, changes in surface elevation have been added to the figures showing segregated ice (see comment above).

[Figure]

*Sum of volumetric water and ice content in the permafrost on the reference date August 31 for the model run* S-clay. *Dark blue colors indicate an increased volumetric water and ice content at the top of the permafrost, where ice segregation takes place. Values in the thawed layer are not displayed. The spin-up covers the years 1860 to 1960. The sum of volumetric water and ice content is shown as soil water can still occur below freezing temperatures dependent on the soil type. For scenario setup see Table 4.*

Furthermore, we changed the figures with the thermal gradients and saturation in the supplement, to clarify our argument. We present here the changes for the gradient in ground temperatures, but the figures for saturation were done in a similar matter.

Supplement, Line 28: *The forcing data set of Bayelva represents* maritime *climatic conditions compared to* the continental setting in Samoylov and therefore *less extreme temperature changes between summer and winter season.* While the *ground temperatures* in the active layer are similar, the permafrost temperatures are much lower in model scenario S-clay compared to B-clay (Fig. S3 and Fig. S4). This leads to higher vertical gradients in ground temperatures at the top of the permafrost in Samoylov, enhancing the formation of segregated ice.

[Figure]

*Ground temperatures on the reference date August 31 for the model scenario* S-clay. *The detailed plot shows the ground temperatures at the permafrost table during the spin-up, where ice segregation takes place.*

[Figure]

*Ground temperatures on the reference date August 31 for the model scenario* B-clay. *The detailed plot shows the ground temperatures at the permafrost table during the spin-up, where ice segregation takes place.*

255

**Ll. 416-417: "In model scenario S-clay-rain50, the permafrost does not degrade to the end of the 21st century and no talik develops as less energy is consumed for thawing and melting of soil water under drier conditions" (underline by the reviewer). I am puzzled how this argument makes a sense.**

This argument was not convincing and a mistake from our side. We have rewritten the part and reasoning should be clear now.

Line 503: *In model scenario S-clay-rain50, the permafrost does not degrade to the end of the 21st century and no talik develops. This can most likely be explained with a lower thermal conductivity of the soil under drier conditions (increasing from 1.04* $\mathrm{W\,m^{-1}K^{-1}}$ *(66 % saturation) to 1.67* $\mathrm{W\,m^{-1}K^{-1}}$ *(100 % saturation)), but other effects such melting of ground ice might play a role.*

**L.419: "1908s" -> "1980s"?**

Yes, this was a typo. It is corrected now.

Line 475: *When the same model set-up is run with forcing data from Bayelva (B-clay), which represents warmer permafrost under moist conditions, we obtain a comparable formation of segregated ice with a slightly lower maximum of 0.028 m in the 1980s compared to 0.034 m with forcing data from Samoylov (S-clay).*

**Figure 7, ll. 418-425: A general feature of Figure 7 looks that the B-clay run produced more segregated ice compared to S-clay run after the 1980s except for the oscillatory decadal periods of comparable segregated ice formation in the 2020s, 2050s, and 2080s. It does not seem a simple warming phenomenon. The argument developed in this paragraph are not well-founded or convincing. Also, it is suspected that the vertical lines showing the end of the spinup periods may be off by 10 years or so if that for the S-runs ends in 1969 and that for the B-run ends in 1989.**

It is true, that the model scenario *B-clay* produces more segregated ice compared to *S-clay* after 1980. This can be explained by the stabilization of the permafrost table over longer time periods (new segregated ice can form). In Samoylov, the active layer deepens faster, so that new segregated ice is melted directly in the following years. We rephrased the paragraph to clarify this for the reader.

The vertical lines showing the end of the spin-up period are correctly placed. While the spin-up for Samoylov repeats the years 1960-1969, the spin-up for Bayelva uses the years 1980-1989 (Table 3). The reason for this is, that we aimed to select a time period, that is most representative for earlier climate conditions at the field site (air temperatures, precipitation etc.). These are different for Samoylov and Bayelva.

Line 481: *During the time period 2020 to 2100, B-clay builds up more segregated ice in deeper soil layers. This can be explained by a slower increase of the active layer thickness compared to Samoylov Island, so that the permafrost table stabilizes for longer time periods, enabling new formation of segregated ice. As a consequence, less ground subsidence takes place for the Bayelva run.*

**Figure 10. The scenario names are different from those found in Table 3 or text.**

We corrected the scenario names in the figure caption.

Figure 12: *Column-accumulated segregated ice and surface elevation changes on the reference date August 31 for the model scenarios S-clay, S-clay-load5 and S-clay-load5-1980. A load that is acting on the soil column from the beginning of the simulation (S-clay-load5) suppresses ice segregation. A load that is added during thaw consolidation (S-clay-load5-1980) accelerates the process and the ground surface subsides faster.*

**Ll. 478-480: I wondered the relative contribution of ice and sedimentation can be quantitatively assessed if the density of ice is constant (and it is apparently assumed so in this module).**

The effective sedimentation rate is about 0.55 m per 1000 years. If the ground shows a subsidence of 0.286 m (*S-clay-sed1000*) between 1980 and 2100, it subsided effectively by 0.286 m in addition to approx. 0.066 m, corresponding to the deposited material during the 120 years.

**Ll. 535-538: It is not clear which figures in supplement B would support the argument.**

We added the figure number in the manuscript.

Line 593: *Moist conditions lead to more segregated ice compared to dry conditions, as more water is available for ice segregation (Fig. S1 in Suppl. S2) and the increased hydraulic conductivity supports the mobilization of soil water. Furthermore, high temperature gradients in the soil column (Fig. S3 in Suppl. S2) enhance ice segregation due to the temperature dependency of the soil matric potential in frozen state (Westermann et al. 2016).*

**Ll. 554-555: "Due to the soil characteristics of the clay, mobile soil water occurs next to the ground ice and is pressed out due to higher total stress" It is not clear whether this statement is meant to be general explanation of the real world, or the result description of the model. Similar ambiguity or confusion was found sporadically in the text.**

Thanks for pointing out this issue. We changed the wording in the manuscript.

Line 614: *Due to the* selected *soil characteristics of the clay* in our model setup (Table 3)*, mobile soil water occurs next to the ground ice and is pressed out due to higher total stress.*

**Ll. 565-566: "Therefore, the time period available for the formation governs today's segregated ice content." This is another example of the above issue.**

We rephrased the sentence.

Line 626: *The shift of the freezing front leads to a thickening of the ice-rich soil layers as demonstrated in this study (Sect. 3.6).* Our results show that longer periods of sedimentation result in thicker ice layers, with the sedimentation rate influencing the overall ice content of the accumulated layers.

**L. 621: It is not clear what "fields sites affected by carbon-rich soils" mean or denote.**

We clarified this statement and changed it in the manuscript.

Line 740: *As these soils are sensitive to changing stress conditions due to a high compression index, it could be beneficial to apply the new model scheme for field sites* with high carbon contents in the soil stratigraphy*.*

**L. 642: It is not clearly stated which part of the manuscript support the claim of being able to "lead to improved simulations of ice-rich ground responding to a warming climate."**

We added a statement in the manuscript.

Line 763: *In this study, we present a new model scheme* for the CryoGrid community model*, which demonstrates ice segregation as well as thaw consolidation.* The model is capable of building up layers of segregated ice with associated ground heave and calculating subsidence upon thawing, which can improve *simulations of ice-rich ground responding to a warming climate.*

**Ll. 650-652: "Climatic conditions, which lead to large gradients…" It is not general "climatic conditions" but warm and moist conditions that derive. It needs to be specific and precise. It is suggested to check the overall manuscript from this perspective.**

360   This is a valid point. We clarified it in this paragraph as well as in other parts of the manuscript.

Line 772: *The model results suggest that several factors play an important role in the formation and melt of segregated ice such as (i) ground temperature gradients and soil water content, (ii) soil type and (iii) external loads. (i) Large gradients in ground temperatures, as well as high soil water contents in the active layer through high precipitation support ice segregation.*

365

Line 374: *All other scenarios are based on a simplified stratigraphy and are designed to analyze different influencing factors: (i) soil water contents and ground temperatures, (ii) soil type (sand, silt, clay and peat), (iii) external loads and (iv) sedimentation.*

370   We changed the title of section 3.3 to *Sensitivity towards soil water content and ground temperatures*

Line 463: *Model simulations with different forcing data sets suggest that ice segregation is highly dependent on the climatic conditions, which can lead to different soil water contents and ground temperatures.*

375   Line 590: *We identify several factors, which influence the soil mechanical processes: (i) soil water contents and ground temperature gradients, (ii) soil type and (iii) external ground loading, which will be discussed in the following.*

---

## Author Comment (AC2)

**Response to anonymous referee 2**

We would like to thank the reviewer for evaluating our manuscript and for the useful comments, which helped to improve it. In the following, we provide a reply to the points discussed by the reviewer as well as changes in the manuscript.

The comments of the reviewer are written in **bold**, the extracts of the manuscript in *italics* with changes highlighted in blue
5 and line numbers referring to the revised manuscript.

**Juditha Aga et al presented a new modeling capability added to CryoGrid model to simulate ice segregation and thaw consolidation considering proof-of-concept scenarios. While the capability is important and of interest to the modeling community of permafrost regions, especially considering ground heave and subsidence evolution under a changing cli-**
10 **mate, I have a few concerns that are listed below.**

**Major comments**

**A better motivation for the study is needed in the Introduction section; why should one care about ice segregation;**
15 **where in the Arctic they form; what observations show us; what has been done already to address ice segregation (not Earth system models but small-scale models). Field imagery (and/or a schematic) of ground heave and subsidence can help set the stage as ground heave and subsidence are linked with ice segregation and thaw consolidation.**

We added the suggestions of the reviewer in the introduction of our manuscript.

We have one paragraph in the introduction, which discusses effects of segregated ice upon thawing, such as geomorphological changes (thermokarst and ground subsidence), a potential contribution to the permafrost carbon feedback and effects on the stability of the ground. We rephrased some parts to improve the paragraph. Furthermore, we included the effect of a delayed permafrost degradation due to ice-enriched soil layers. We mention this also as a missing element in previous model studies.
25

Line 50: *The ground ice content and its distribution strongly determines the sensitivity of permafrost to thaw (Jorgenson et al., 2010; Nitzbon et al., 2019). Ice-rich layers in the soil can delay permafrost degradation as energy is consumed upon melting of the ground ice, which is consequently not available for the warming of the ground. In addition, ice segregation can continue even under thawing conditions, forming ice layers at the top of the permafrost, continuously delaying the warming*
30 *process. However, if excess ice is melted, it can result in thermokarst and ground subsidence (Farquharson et al., 2019; Kokelj and Jorgenson, 2013; Nitzbon et al., 2019). In consequence, substantial geomorphological changes reshape the landscape, manifested in the formation of lakes, thaw slumps, gullies and the transformation of low-centered to high-centered polygons (Kokelj and Jorgenson, 2013; Liljedahl et al., 2016; Nitzbon et al., 2019). These processes could contribute to accelerating the*

*mobilization of permafrost carbon, which may further increase atmospheric carbon concentrations, a process known as the* permafrost carbon feedback *(Miner et al., 2022; Schuur et al., 2008). Furthermore, ground ice controls the* hydrological *and mechanical properties of the soil by reducing permeability and increasing the mechanical strength (Painter and Karra, 2014). These parameters control the structural stability of the ground. Upon thawing, mass movements along slopes might increase and together with ground subsidence, the reduced stability can endanger human infrastructure and settlements (Hjort et al., 2022; Schneider von Deimling et al., 2021).*

Line 75: *As a consequence, previous models targeting ground ice thaw and thermokarst require excess ice distributions prescribed from field observations, which makes applications at sites without ground ice data challenging. Furthermore, as ice segregation is not implemented, they neglect the delay in permafrost warming through formation of new segregated ice layers during the simulation period.*

We added a new paragraph the occurrence of segregated ice.

Line 41: *Layers of segregated ice in the ground are widespread in permafrost environments, especially in fine-grained sediments, which are susceptible to ice segregation (French and Shur, 2010). Cryostratigraphic mapping has been performed in numerous studies, documenting segregated ice especially in Siberia (Andreev et al., 2009; Meyer et al., 2002; Schirrmeister et al., 2008; Siegert and Babiy, 2002) and North America (French et al., 1986; Heginbottom, 1995; Kanevskiy et al., 2013; Shur and Jorgenson, 1998). O'Neill et al. (2019) modelled the occurrence of segregated ice in Canada, showing abundance in fine-grained lacustrine sediments, raised peat plateaus and uplifted marine sediments.*

We added a new paragraph in the manuscript about what observations can show us.

Line 46: *Observations of the cryostructure, including segregated ice, can reveal information about the evolution of the ground, e.g. if the permafrost was formed syngenetically or epigenetically. Furthermore, thaw unconformities, such as former active layers, can be detected by changes in the ice content with depth and the isotopic signature of the ground ice (French and Shur, 2010).*

We extended the references that we already mentioned with a very interesting study that demonstrates palsa formation through ice segregation.

Line 69: *Thaw consolidation has been the focus of model development for many decades (Morgenstern and Nixon, 1971; Nixon and Morgenstern, 1973; Sykes et al., 1974; Foriero and Ladanyi, 1995; Dumais and Konrad, 2018), and different approaches have been presented for modelling ice segregation (Fu et al., 2022; Fisher et al., 2020; Lacelle et al., 2022). An example is the approach of An and Allard (1995), who successfully demonstrated palsa formation through accumulation of*

*segregation ice. However, these processes are typically not implemented in land surface models and simulating the long-term*
70   *evolution of segregated ice has not been performed yet (Fu et al., 2022).*

We added a schematic of ice segregation and ground heave in the introduction to visualize the process for the reader.

[Figure]

*Illustration of ground heave through ice segregation at the base of the active layer. (a) Lenses of segregated ice are formed*
75   *at the top of the permafrost. (b) If the segregated ice is preserved over a long time period, layers with segregated (excess) ice*
*are forming, causing heave of the ground surface. Figure modified after Fu et al. (2022).*

**Lots of details have been provided about the model in the Methods but they are hard to follow, especially the description of CryoGrid class (modules) needs a better workflow; a schematic would be helpful to understand what parts of**
80   **the existing CryoGrid have been modified and what new parts are added.**

We rephrased extensive parts of the description of CryoGrid to improve the clarity. Furthermore, we added a table with an overview with the model components and how they have been changed.

85   Line 91: *In this work, we extend the capabilities of the CryoGrid community model (Westermann et al., 2023) with a representation of soil mechanical processes. The CryoGrid community model is a modular framework for simulating the permafrost thermal state and the water and ice balance. To set up simulations, the user can choose between different so-called "stratigraphy classes", which are characterized by specific model physics and state variables. As an example, one stratigraphy class can calculate soil water contents with a simple model, while another can account for water redistribution through Richards*
90   *equation in unsaturated soils. Furthermore, there are dedicated stratigraphy classes for non-ground materials, in particular*

*for the seasonal snow cover. The stratigraphy classes can be stacked vertically, so that the available classes representing snow can can be flexibly combined with a range of classes for ground materials.*

*In this study, we demonstrate a new stratigraphy class denoted GROUND_freezeC_RichardsEq_seb_pressure, a fully fledged process model for soils. It is based on the already existing stratigraphy class GROUND_freezeC_RichardsEq_seb (Westermann*
95 *et al., 2023) and inherits many of its functionalities. While a detailed description of the CryoGrid community model is provided in Westermann et al. (2023), we summarize the main aspects relevant for this work in Sect. 2.1, before describing the defining equations and main properties of the new stratigraphy class. The model is demonstrated and evaluated for two field sites (Sect. 2.2), for which we simulate a range of model scenarios (Sect. 2.4) with different settings for subsurface properties and model parameters (Sect. 2.3).*

100

*Line 107: In the stratigraphy classs GROUND_freezeC_RichardsEq_seb, each model grid cell is characterized by its volumetric contents of the mineral, organic, water and ice components, which also define the porosity. The upper boundary is defined as the interface between the ground surface and the atmosphere at which the surface energy balance is applied, controlled by the exchange of short-wave and long-wave radiation, as well as latent and sensible heat fluxes. To calculate the*
105 *surface energy balance, the forcing data must prescribe time series of air temperature, solid and liquid precipitation, wind speed, short-wave and long-wave radiation, specific humidity and air pressure. The lower boundary condition (set at a user-specified depth) is defined by a constant geothermal heat flux.*

*Subsurface heat transfer is based on both conductive and advective fluxes. The calculation of heat conduction follows Fourier's law with the thermal conductivity of the material being the controlling factor. The heat transfer through advection is*
110 *determined by vertical water fluxes.*

*A soil freezing characteristics describing the relationship between ground temperature and unfrozen water content (Painter and Karra, 2014) is implemented in GROUND_freezeC_RichardsEq_seb. To determine liquid water and ice contents in frozen soils, we first calculate the matric potential for unfrozen conditions $\psi_0$ [m] and based on that, the matric potential in frozen state $\psi$ [m] from which the water content can be inferred (assuming no residual water). A detailed description of the approach*
115 *can be found in Westermann et al. (2023).*

*Table 1. Different model components, their implementation in the CryoGrid community model after Westermann et al. (2023) and the additions to the model code, presented in this study.*

| Model component | Base class GROUND_freezeC_RichardsEq_seb | Additions in GROUND_freezeC_RichardsEq_seb_pressure |
|---|---|---|
| Upper boundary condition | Surface energy balance | - |
| Lower boundary condition | Geothermal heat flux | - |
| Subsurface heat transfer | Heat conduction and heat convection | - |
| Soil freezing characteristics | after Painter and Karra (2014) | - |
| Water flow in unsaturated conditions | Richard's equation | - |
| Water flow in saturated conditions | Gravity-driven | Additional water flow when excess pore water pressures occur |
| Lateral water transport | Overland flow | - |
| Stress conditions in the ground | - | Calculation of total and effective stresses |
| Compressibility of the soil column | - | Calculated from the compression curve |
| Excess ground ice | Defined in the initial conditions, melting of excess ice possible | Formation and melt of segregated ice |
| Sedimentation | - | Material is added with user-defined properties |

**The manuscript mainly focused on Samoylov (S) scenarios and not Bayelva (showed just one B-clay). If the focus is**
120 **more on one site, it would make sense to drop the other to not confuse the readers. Samoylov's climate can be made synthetically warmer as the authors are performing proof-of-concept simulations and not validating the model.**

We added two more model scenario for the Bayelva field site (*B-clay-sed1000* and *B-clay-sed370-2x*), so that the manuscript is no longer entirely focused on Samoylov. Furthermore, we compare the results of this model scenario with in situ observations
125 from a site with high-quality ground ice observations on Svalbard (see comment below).

CryoGrid uses the surface energy balance as the upper boundary, so that it is not possible to simply make it warmer, as warming in reality also changes other parts of the forcing, in particular the incoming radiation and humidity. Therefore, Bayelva needs to be included in this study.

130

**While I personally believe projections are not needed here to demonstrate the capability, since they are there how do they compare against simulation without the formation of ice segregation?**

We performed a model run without formation of segregated ice and included the analysis in the supplement.

Supplement, Line 43: *S4: Control runs without formation of segregated ice*

*We perform a model run based on the peat stratigraphy (Table 3) without ice segregation (S-peat-control). To do so, we suppress water flow into already saturated and frozen grid cells, while the rest of the functionality of the model stays the same.*

*The model scenario S-peat forms segregated ice predominantly between 1861 and 2000 (Sect. 3.4). During this time period, the active layer thickness is on average $0.054 \pm 0.029$ m shallower for S-peat compared to S-peat-control. As S-peat forms segregated ice at the top of the permafrost each year, this excess ice melts partly in the following year, consuming energy, which is consequently not available for warming of the ground. As S-peat-control does not contain segregated ice, the energy can be used directly to warm the ground, resulting in a deeper active layer.*

*In the time period 2000 to 2010, large parts of the segregated ice are melted in S-peat (Sect. 3.4). Again, this process requires energy, even more than during 1861 to 2000, as more ground ice is melted during these years. Therefore, the difference in active layer thickness increases to $0.078 \pm 0.088$ m on average, with S-peat having the shallower active layer than S-peat-control.*

*From 2010 to 2100, S-peat forms less segregated ice, however, ice segregation continues at the top of the permafrost on a smaller scale (Sect. 3.4). Therefore, a shallower active layer is still simulated for S-peat compared to S-peat-control, even though the difference in active layer thickness decreases to $0.039 \pm 0.053$ m on average.*

*The comparison between S-peat-control and the model scenario S-peat with ice segregation shows, that the formation of segregated ice leads to shallower active layers, especially during the periods where the ice-enriched soil layer thaws. This can be explained by the energy required for ground ice melt, which is then not available for ground warming.*

**I have also some reservations about the reference date of August 31 when using different climates, this might be fall freeze-up time for one site but not for another site. So, the comparison that on August 31 one site showed this much ice segregation and another showed that much is not very convincing.**

We checked the validity of the reference date for both field sites by comparing the vol. water and ice content at the upper 20 cm and 1 m of the permafrost of the reference date August 31 and the day with the maximum active layer thickness for each year. We calculated the following mean differences with corresponding standard deviations:

Samoylov: $8.4e-05 \pm 0.0269$ (20 cm) and $-0.0012 \pm 0.0107$ (1 m)

Bayelva: $-0.0050 \pm 0.0202$ (20 cm) and $-0.0020 \pm 0.0072$ (1 m)

We are therefore confident, that we can use the results August 31 for our analysis. A further advantage of using a fixed date rather than the state at maximum thaw is that we can use the same procedure when a talik has formed. In this case, we would

need to introduce additional criteria on how the point in time was selected after a talik has formed, which makes it more complicated to compare ground ice contents throughout the figure. We added a statement in the revised version of the manuscript.

Line 407: *In the following, we present results of the model scenarios described in Sect. 2.4 and summarized in Table 4. To ensure comparability between the different sites and years, results are always provided at August 31.*

**I would suggest just focusing on the historical climate (1000s of years) and showing how thick ice lenses (segregated ice) formed in the past – at least show the simulated segregated ice is in some comparison with the field observations. Showing the formation of 2-3 cm segregated ice is not very appealing given it is a modeling paper and the main focus is ice segregation and thaw consolidation, so the authors need to explore it further.**

It is true, that the formation of thick segregated ice layers takes typically 1000s of years. To realistically simulate this process, we would require long-term historical forcing data. As we do not yet have such historical forcing data available at this point, we use a model spin-up by repeating a 10-years period. The model scenarios with a spin-up of 1000 years show, that the model is capable of building up thick ice lenses through ice segregation, while the model scenarios with a spin-up of 100 years investigate the sensitivity to different influencing factors. As the aim of the manuscript is to demonstrate the functionality of the model, we are confident that the chosen spin-up periods support the objectives of the study. To simulate a the formation of segregated ice at a specific field site, long-term historical forcing data would be beneficial as the reviewer suggests. Such simulations would be in principle possible with the presented model, but preparing bias-free historical forcing data is challenging and beyond the scope of this work. However, it would be an interesting aspect for future studies.

Although a comparison with field measurements is challenging due to lacking historical forcing data, we added a comparison with in situ observations of ground ice contents which indeed suggest that we can produce results in the right order of magnitude. Concretely, we added a sedimentation run with increased sedimentation rates for Samoylov (*S-clay-sed350-3x*) to compare it to Nitzbon et al. (2019). Furthermore, we set up a model scenario with increased sedimentation rates as well for Bayelva (*B-clay-sed370-2x*) and compare it to the study of Cable et al. (2018). We included the runs in the results and added a new section to the discussion, which is shown below. We are grateful to the reviewer for this valuable suggestion which in our opinion has improved the quality of the study!

[revised manuscript text omitted]

**Minor comments**

**L2. What does "very ice-rich soils" mean?**

255

We rephrased the sentence.

Line 2: *The ground ice content in cold environments influences the permafrost thermal regime and the thaw trajectories in a*
*warming climate, especially for soils containing excess ice.*

260

**L24: Sibiria should be Siberia**

Fair enough. We corrected it.

265    Line 24: *Especially in Siberia and Alaska, they are often associated with a high content in excess ground ice.*

**L2531-34: the formation of segregation ice needs a clear description. Cryosuction can even happen from top to bottom.**

We added the definition from the "glossary of permafrost and related ground-ice terms" (Harris et al., 1998) for a clear description. We mention in this paragraph, that the water can be drawn from the active layer towards the freezing front, i.e. from top to bottom.

Line 31: *Segregated ice is ground ice, which forms through migration of soil water to the frozen fringe (Harris et al., 1998). It occurs as discrete ice lenses or layers, which can range from less than a millimeter to more than 10 m (French and Shur, 2010; Harris et al., 1988). Segregated ice is typically associated with ground heave as the ice content in the ground increases (Fig. 1; Miller, 1972; Taber, 1929). It can build up in epigenetic permafrost when unfrozen soil water from the active layer is attracted towards the freezing front (Guodong, 1983; Mackay, 1983; Taber, 1929), accumulating ice near the permafrost table. Besides, ice segregation can occur at the base of the permafrost, as water is drawn from the soil below as permafrost is forming. These ice-enriched layers are widespread in permafrost environments (French and Shur, 2010) as for example in Canada (O'Neill et al., 2019). However, the evolution of syngenetic permafrost can result in enhanced ice formation. Accumulation of organic material as well as sedimentation in alluvial, eolian or hillslope settings can lead to a rise in the permafrost table and hence a growth of segregated ice (Guodong, 1983; French and Shur, 2010). In this context, segregated ice can form also together with syngenetic ice wedge growth, forming ice lenses within polygonal permafrost as observed in Yedoma deposits (Schirrmeister et al., 2013).*

**L43: "thermokarst and subsequent ground subsidence" does it mean thermokarst happens first and ground subsides afterward? What is thermokarst?**

We removed "subsequent" from the text.

Line 53: *However, if excess ice is melted, it can result in thermokarst and  ground subsidence.*

**L71: "contrasting permafrost sites" in terms of what?**

We specified it in the revised manuscript.

Line 85: *To do so, we run various model scenarios with climate data from two contrasting permafrost sites, representing cold continental and relatively warm maritime permafrost conditions.*

**L94: relative humidity or specific humidity? Please specify**

We clarified it in the manuscript.

Line 110: *To calculate the surface energy balance, the forcing data must prescribe time series of air temperature, solid and liquid precipitation, wind speed, short-wave and long-wave radiation, specific humidity and air pressure.*

**Section 2.1 Model description. A schematic here can really help understand all the processes in the model and how these processes are linked together. While the authors have explained the model in the text, it is still confusing for those who are not very familiar with the model.**

We added a table in section 2.1, explaining where changes has been made to the model code of the CryoGrid community model. Furthermore, we rephrased large parts o the CryoGrid community model description for clarity. For details, see reply on the corresponding major comment.

**L106: define the sub/superscripts.**

We defined the sub- and superscripts.

Line 122: *The water balance in* GROUND_freezeC_RichardsEq_seb *is based on vertical water flow $j_w^v$ [$\mathrm{m\,s^{-1}}$] controlled by the Richards equation (Richards, 1931):*

$$j_w^v = -K_w \left( \frac{\partial \psi}{\partial z} + 1 \right) \tag{1}$$

*with $\psi$ [$\mathrm{m}$] being the matric potential, $z$ [$\mathrm{m}$] the vertical coordinate and $K_w$ [$\mathrm{m\,s^{-1}}$] the hydraulic conductivity. The subscript $w$ denotes water and the superscript $v$ signifies vertical for model variables.*

**L128: "table" should be Table**

We corrected it in the manuscript.

Line 149: *To do so, a set of additional state variables is necessary, which can be found in Table 2.*

**L168: This is a model development paper, instead of referring the reader somewhere else please provide the definition here.**

335    We added a statement why we used the bulk quantities in the model instead of referring only to Westermann et al. (2023).

Line 192: *with $\Theta_m$ [m³] and $\Theta_o$ [m³] being the bulk volumetric content of the mineral and organic components, i.e. the absolute volume in a grid cell filled by the respective component. In GROUND_freezeC_RichardsEq_seb, bulk quantities as $\Theta_m$ and $\Theta_o$ are conveniently used as state variables (Westermann et al., 2023), so that the grid cell thickness d for unsaturated*
340    *grid cells can be updated with the porosity obtained from Eq. 7 (see Sect. 2.1.2 for the saturated case).*

**L247: So Bayelva is warm and maritime, and Samoylov is cold and continental? The climate is mentioned for one site but not for the other.**

345    We changed the manuscript accordingly.

Line 274: *Samoylov island (located in northern Siberia) represents a continental setting with cold continuous permafrost, while the Bayelva field site (located on Svalbard) is characterized by a maritime climate with warm, but still continuous permafrost.*

350

**L251: "In summer, precipitation and evapotranspiration balance each other." It is not clear if this statement is only for this particular site or not. Please clarify. I am not sure if this is true across the Arctic, it depends on the climate.**

We clarified, that this statement is only true for the Samoylov field site.

355

Line 280: *In summer, precipitation and evapotranspiration balance each other in Samoylov.*

**L255-265: many abbreviations without definitions.**

360    We added the definitions for the abbreviations.

Line 289: *We use the same forcing data set as Westermann et al. (2016), Nitzbon et al. (2019) and Nitzbon et al. (2020) for the long-term thaw susceptibility run in that study. The climatic forcing between 1960 and 2012 is derived from the reanalysis product CRU-NCEP combining data from the Climate Research Unit and National Centers for Environmental Prediction, see*
365    *Kalnay et al., 1996; Harris et al., 2014), downscaled for Samoylov with in situ data from the automatic weather station. After 2012, the forcing data is taken from the Community Climate System Model CCSM4 outputs simulated under the Representative*

*Concentration Pathway (RCP) 8.5 scenario (Meehl et al., 2012).*

**L267-268: 870 mm rainfall and 668 mm snow, so the total is about 1500 m annual precipitation. Is this for Samoylov? This is a lot different than what other studies have reported for example Liljedahl et al. (2016)**

We thank the reviewer for pointing out this inconsistency. We checked our forcing data, and we made some mistake by calculating the annual precipitation (we took mm instead of mm/d). The forcing data itself is correct, so that the simulations are not affected. We corrected the numbers in the manuscript.

Line 297: *An increase in air temperature from -16.7 °C to -8.1 °C, an increase in longwave radiation from 214 to 249 W m$^{-2}$, a slight decrease in shortwave radiation from 105 to 101 W m$^{-2}$, a pronounced increase in rainfall from 157 to 218 mm and an increase in snowfall from 133 to 167 mm.*

**L298: "undecomposed organic material features coarse pores" how old is this organic material? Also, is this just an assumption or the organic material is undecomposed at those sites?**

The layer in the upper 15 cm represents the moss layer on top and is therefore poorly decomposed as fresh vegetation is still growing here. We rephrased the sentence to clarify this.

Line 330: *Hereby, we use the same soil properties in the upper 0.15 m as in the multi-layer stratigraphy to account for the insulating effect of the organic-rich moss layer on top. The poorly decomposed organic material features coarse pores with unknown values for the van Genuchten parameters $n$ and $\alpha$.*

**L317: 1100 m: +10.2C, does this deep soil temperature also come from borehole measurements?**

No, this value comes from a steady-state temperature profile corresponding to a geothermal heat flux of 50 mWm$^{-2}$. This is described in the manuscript.

Line 351: *For Samoylov, we used the same initial ground temperatures as Nitzbon et al. (2019), as we used the same forcing data as this study: 0 m depth: 0.0°C; 2 m: -2.0°C; 5 m: -7.0°C; 10 m: -9.0°C; 25 m: -9.0°C; 100 m: -8.0°C; 1100 m: +10.2°C. These values are based on borehole data from 2006 and a steady-state temperature profile corresponding to a geothermal heat flux of 50 mWm$^{-2}$.*

400 **L320: what is the total depth of the soil column, Table 2 lists properties for 0-9 m but L317 provides soil temperatures with depths of 0-1100 m.**

We thank the reviewer for pointing out that this was not clear and we changed the manuscript accordingly. The entire model domain is 100 m, soil temperatures in greater depths are given for interpolation of the ground temperatures in greater depths.
405 Table 2 lists only the characteristics of the upper 9 m, as the new stratigraphy class is applied here.

Line 317: *We use a model domain reaching from the surface to 100 m depth, which is described by a stack of two different ground classes. Below 9 m depth, soil mechanical processes are not accounted for due to frozen conditions and high total stress during the entire simulation. Therefore, we apply the stratigraphy class* GROUND_freeW_seb*, which operates without*
410 *soil mechanical processes and water balance, as described in Westermann et al. (2023). Between the ground surface and 9 m depth, we use the new stratigraphy class GROUND_freezeC_RichardsEq_seb_pressure as presented in this study.*

Caption of table 3: *Stratigraphies used for the model scenarios in this study in the upper 9 m of the model domain.*

415 Line 361: *The initial vertical resolution of the grid cells increases stepwise from the surface to greater depths (0-1 m depth: 0.05 m; 1-5 m depth: 0.1 m; 5-10 m depth: 0.2 m; 10-20 m depth: 0.5 m; 20-50 m depth: 1 m; 50-100 m depth: 5 m). The fine resolution in the top layers allows us to analyze the ground temperatures in detail in the upper soil layers.*

**L318: where did this geothermal value come from? Any reference?**
420

We added the reference.

Line 353: *These values are based on borehole data from 2006 and a steady-state temperature profile corresponding to a geothermal heat flux of 50* $\mathrm{mW\,m^{-2}}$ *(Langer et al., 2013).*
425

**Table 2: what is the residual saturation?**

We assume no residual water as Westermann et al. (2023).

430 Line 119: *[...]we first calculate the matric potential for unfrozen conditions* $\psi_0$ *[m] from which the matric potential in a frozen state* $\psi$ *[m] and finally the water content can be inferred (assuming no residual water).*

Caption of Table 3: *We assume no residual water.*

**L330-335: This is totally confusing, the authors mentioned considering the undecomposed organic matter and now they started talking about peat which has totally different hydraulic and thermal properties. Please explain whether are you still using sandy soil properties for the organic layer and how this part (L330 onwards) is related to L298. I would assume most of the organic material at Samoylov is (partially)decomposed.**

The upper 0.15 m represent the moss layer on top and is applied for each model run. In contrast, the organic soil in model scenario *S-peat* represents peat (0.15-9 m depth), which has different characteristics than the moss layer (Table 3). We clarified this in the manuscript.

Line 330: *Hereby, we use the same soil properties in the upper 0.15 m as in the multi-layer stratigraphy to account for the insulating effect of the organic-rich moss layer on top. The poorly decomposed organic material features coarse pores with unknown values for the van Genuchten parameters n and α. Therefore, it is phenomenologically set to the properties of coarse-grained (sandy) soil, which has a broadly similar retention characteristic. Between 0.15 and 9 m depth, we set homogeneous soil properties of one soil type, distinguishing between sand, silt, clay and peat. They feature different characteristics for volumetric fractions of mineral and organic content, saturation, initial void ratio, residual stress, compression index, permeability, α coefficient and n coefficient. The chosen values for the different settings can be found in Table 3.*

**L342: "effect of different soil types" on what?**

We added this in the manuscript.

Line 383: *A comparison of the model runs S-sand, S-silt, S-clay and S-peat show the effect of different soil types on ice segregation and thaw consolidation.*

**Table 3. If out of 12, 11 scenarios are for the Samoylov site then what is the purpose of including Bayelva site? Both sites should be treated similarly or just remove Bayelva site from the manuscript and focus on one site.**

The Bayelva site is important to analyse the difference between a maritime and continental setting. Therefore, the forcing data of Samoylov cannot be made synthetically warmer (see reply to major comment above). Instead of removing Bayelva from the manuscript, we added two more model scenarios at the Bayelva field site.

**L365: Polygon centers are not representative of the entire polygon. Active layer thickness varies among troughs, rims, and centers. Although ALT varies across microtopographic locations, it would be reasonable to compare it against the average ALT (average of rims, troughs, and centers).**

470    We explain in the revised version of the manuscript, why we only compare our model results against polygon centers.

   Line 411: *For the Samoylov island site, we perform model validation focusing on polygon centers where segregated ice is normally found. In detail, we compare the validation run S-val designed to represent the typical ground stratigraphy of polygon centers at the site (Sects. 2.3, 2.4) to published in situ temperature data from 0.05 m and 0.40 m depth at a polygon center near*
475 *the northern shore of Samoylov Island* (Boike et al., 2013; Langer et al., 2011a, b; Westermann et al., 2016).

   **L370-374: Have you looked at the comparison of the observed and simulated snow depths to support your reasoning for the mismatch? Have you compared the forcing data to any nearby climate station? Also, since you have taken the observed soil temperature data from a polygon center, was the polygon center inundated during the fall freeze-up? So**
480 **it is not just the forcing (snow depth) that could affect freeze-up.**

   For our simulations, we limit the snow depth to 45 cm, following Westermann et al. (2016), who reasons this choice as following:
   "In addition, the maximum snow depth in CryoGrid 3 is restricted to 0.45 m, the approximate height of the polygon rims
485 above the centers. Snow depth observations of Boike et al. (2013) show that the maximum height in polygon centers rarely exceeds these values, since further snowfall is largely blown away by consistently strong winds."
   We explain this in our manuscript:

   Line 395: *For Samoylov, we set a threshold for the snow depth to account for observed snow ablation due to wind drift*
490 *following the approach of Westermann et al. (2016). As the snow in the polygon centers is protected by the surrounding rims, we set a value of 0.45 m, which is in line with measured snow depths in polygons centers on Samoylov Island (Boike et al., 2013).*

   For the validation run, we have temperature data from a polygon available, which is partially disintegrated with lower poly-
495 gon rims (Boike et al., 2013), leading to overall lower snow depths there. This information is given in the manuscript:

   Line 398: *For validation, we use temperature measurements in a polygon center. However, these measurements are not conducted in the middle of Samoylov island but close to the edge of the island where the wind drift is stronger and the height of the rims above the polygon center is reduced. Therefore, we reduce the maximum snow depth for the validation run to 0.20 m.*
500
   Boike et al. (2013) does not describe inundation of the polygon centers.

   **L382: this is confusing again, at L317 it says -7C at 5 m depth, here is it -9C at 5 m depth. Why is that?**

505     As we used the same forcing data as Nitzbon et al. (2020), we used the same initial ground temperatures to be consistent. We conduct long-term simulations and the temperature profile in the uppermost meters is independent of the initialization after a few years, so that this should not influence the model results. We state this in the mansucript:

    Line 351: *For Samoylov, we used the same initial ground temperatures as Nitzbon et al. (2019), as we used the same forcing*
510 *data as this study.*

    Line 356: *Since we conduct simulations on at least centennial timescales and the temperature profile in the uppermost me-*
*ters becomes independent in initialization after a few years, the initial temperatures do not affect simulation results.*

515 **Figure 5: How do these results differ from those of Nitzbon et al. (2020)? Other than you not considering the ice-rich**
**zone**

    Nitzbon et al. (2020) conducted coupled simulations, taking into account the 3D nature of the ground ice distribution in polygonal terrain, which was found to strongly influence the model results. Therefore, the results cannot be compared to 1D
520 simulations here, so that we compare to Westermann et al. (2016), who present 1D simulations for Samoylov with the same forcing data. To do so, we changed figure 5, so that it shows mean annual ground temperatures at 10 m depth as in Westermann et al. (2016). Figure 5 was changed as following:

[Figure]

    *Mean annual ground temperatures $MAGT$ in 10 m depth, depth of the top of the permafrost and the seasonal frost in the*
525 *model scenario S-clay. The spin-up covers the years 1860 to 1960. $MAGT$ warm from around -9°C until the 1980s to values*
*close to the thaw threshold at the end of the 21st century. The depth of the permafrost table deepens from around 0.7 m to 5 m,*
*so that the seasonal frost doesn't reach the same depth towards the end of the century and a talik develops.*

Line 436: *The modelled changes in annual ground temperatures are in the same range as simulation results for the Samoylov field site in Westermann et al. (2016), as pointed out by the following comparison of approximate ground temperatures in 10 m depth (data of Westermann et al. (2016) shown in brackets): -9 °C (-10 °C) during the spin-up, -5 °C (-5 °C) in 2040 and -1 °C (-1 °C) in 2090.*

**Figure 6: What is the focus of this figure, ice segregation or ALT or talik? The section heading says Formation of ice segregation, but the ice segregation and its formation are hard to see (visually) which makes it difficult to understand what is going on. A better comparison here would be to compare simulations with and without the formation of ice segregation and how much will it affect project ALT.**

The focus of this figure should be the ice segregation, i.e. the volumetric water and ice content in the permafrost. The colours indicate, where segregated ice is formed, which is visualized with the dark blue colours at the top of the permafrost. To clarify this, we added an additional statement in the caption. The same was done for figure 9 (now figure 10) and figure 11 (now figure 12).

Caption figure 8: *Sum of volumetric water and ice content in the permafrost on the reference date August 31 for the model run S-clay.* *Dark blue colors indicate an increased volumetric water and ice content at the top of the permafrost, where ice segregation takes place.* *Values in the thawed layer are not displayed. The spin-up covers the years 1860 to 1960.* *The sum of volumetric water and ice content is shown as soil water can still occur below freezing temperatures dependent on the soil type. For scenario setup see* *Table 4.*

We agree that it is hard to see visually the ground heave due to ice segregation. We therefore added a plot with surface elevation change to the figures 8, 9 and 11 (numbering in revised version) and discussed it in the text.

Furthermore, we run a comparison with and without ice segregation. Please check the reply on the major comment in the beginning of this document.

**L421-423: this contradicts the formation of more segregated ice in the future. Figure 7 does show that B-clay leads to more segregated ice.**

We rephrased the paragraph to clarify, that the higher amount of segregated ice in the *B-clay* scenario can be explained by the stabilization of the permafrost table.

Line 481: *During the time period 2020 to 2100,* B-clay builds up more segregated ice in deeper soil layers. *This can be explained by a slower increase of* the active layer *thickness compared to Samoylov Island, so that the permafrost table* stabilizes

*for longer time periods, enabling new formation of segregated ice. As a consequence, less ground subsidence takes place for*
565   *the Bayelva run.*

**In section 2.2, the authors talked about different simulations, but it is not clear what type of simulations are planned in this work. Please clearly state the set of simulations or summarize them in a table. A simulation description is lacking. It would help better follow the description if this section is split into field sites and forcing data.**

570

Section 2.2 gives an overview about the field sites. We included the forcing data in this section for a better text flow, as the manuscript already contains lots of subsections.

The manuscript contains Tables 3 and 4, which summarize the different scenarios including details on stratigraphies, model
575   period and model forcing. Section 2.4 describes the model scenarios. We added a link to this in section 2.2, so that the reader can see easily, where to look up information about the model scenarios.

Line 273: *To demonstrate the capabilities of the new model, we perform sensitivity studies (Sect. 2.4) for two different permafrost sites with strongly different climate conditions*

580

**Section 3.4. Table 2 shows that the top 15 cm has identical soil properties. What is actually the S-peat scenario? How does it differ in terms of peat thickness in the soil from other scenarios (soil types)? Peat has different thermal and hydraulic properties (L344) than other soil types and leads to shallower ALT, but the thickness of the peat is important. If in summers, the peat keeps the ALT colder than in winters it helps prevent escaping heat from the soil. I**
585   **am not sure if the entire soil column is peat in the S-peat scenario. If the entire 9m (or so) column has peat, is it realistic?**

The *S-peat* scenario contains the moss layer on the top as all other scenarios. Below, it consists of peat until 9 m depth. The characteristics are given in Table 3. We agree with the reviewer, that the thickness of the peat is important, when representing a specific field site. However, we do not aim to represent Samoylov with the *S-peat* stratigraphy, but it is part of a sensitivity
590   study, which should show the effect of different soil types on ice segragation and thaw consolidation. To do so, we simplify the stratigraphy to one soil type, similar with sand, silt and clay. We explain the concept of the sensitivity study in the manuscript. Furthermore, we clarify the *S-peat* scenario in the revised manuscript. The soil characteristics of peat are given in Table 2.

Line 329: *To analyze the performance and sensitivity of our model for Bayelva and Samoylov, we used simplified strati-*
595   *graphies to provide standardized and comparable model setups. Hereby, we use the same soil properties in the upper 0.15 m as in the multi-layer stratigraphy from Samoylov island (see above) to account for the insulating effect of the organic-rich moss layer on top. The poorly decomposed organic material features coarse pores with unknown values for the van Genuchten parameters $n$ and $\alpha$. Therefore, it is phenomenologically set to the properties of coarse-grained (sandy) soil, which has a*

*broadly similar retention characteristic. Between 0.15 and 9 m depth, we set homogeneous soil properties of one soil type,*

600  *distinguishing between sand, silt, clay and peat. They feature different characteristics for volumetric fractions of mineral and organic content, saturation, initial void ratio, residual stress, compression index, permeability, α coefficient and n coefficient. The chosen values for the different settings can be found in Table 3.*

**L586: "complex lateral processes"? Unless you are considering 3D simulations at a larger scale, which are mostly not**

605  **feasible for permafrost regions, what complexity can lateral processes bring? My guess is even using CryoGrid with three tiles should not make it complex, but just an additional process.**

We agree with the reviewer and removed "complex" from the manuscript.

610  Line 704: *In its present form, the model is one-dimensional and hence does not account for*  *lateral processes, e.g. lateral water flow due to cryosuction.*

**Check for typos**

615  We checked the entire manuscript for typos and corrected them.

**Map showing the location of the field sites is missing**

We added a map showing the two field sites.

620

[Figure]

*Figure 5: Location of Samoylov Island in the Lena River delta in northeastern Siberia and of the Bayelva climate station on Svalbard. The aerial images of the surroundings of the field sites are shown in the lower left for Samoylov Island and on the lower right of the Bayelva climate station.*

---

## Author Response (AR2)

**Letter to the editor**

We would like to thank both reviewers for their thoughtful suggestions and the editor for handling our manuscript. We prepared a revised version in which we addressed all comments of both reviewers. Here, we give an overview of the main changes to our manuscript, as well as a point-to-point reply with line numbers referring to the revised manuscript.

**Main changes to the manuscript**

1. Reviewer 2: We modified Figure 1 as suggested by the reviewer.

2. Reviewer 3: We extended a paragraph in Sect. 4.4 (Comparison to in situ observations of ground ice) and added a new paragraph in Sect. 4.5 (Limitations) to discuss the modelling of ice segregation during permafrost formation.

3. Reviewer 3: We included the suggested references.

**Response to anonymous referee 2**

We would like to thank the reviewer for evaluating our answers to the review and accepting the majority of our revisions. In the following, we provide a reply to the remaining points discussed by the reviewer as well as changes in the manuscript.

The comments of the reviewer are written in **bold**, the extracts of the manuscript in *italics* with changes highlighted in blue and line numbers referring to the revised manuscript.

**I am happy with most of the revisions made. Although I have one comment about the schematic in Figure 1. Why is the top of the segregated ice flat (visually that's what it appears)? Conceptually, the surface topography should follow the topography of the segregated ice (see, for example, Figure 2 at https://tc.copernicus.org/articles/13/97/2019/)**

We thank the reviewer for the feedback. We changed the figure so that the surface topography follows the topography of the segregated ice. Furthermore, it shows the formation of segregated ice at the base of the permafrost, which was suggested by another reviewer.

[Figure]

**Figure 1.** Illustration of ground heave through ice segregation at the top and the base of the permafrost. (a) Lenses of segregated ice are formed. (b) If the segregated ice is preserved over a long time period, layers with segregated (excess) ice are forming, causing heave of the ground surface. Figure modified after Fu et al. (2022).

**Response to anonymous referee 3**

We would like to thank the reviewer for evaluating our manuscript and for the useful comments, which helped to improve it. In the following, we provide a reply to the points discussed by the reviewer as well as changes in the manuscript.

The comments of the reviewer are written in **bold**, the extracts of the manuscript in *italics* with changes highlighted in blue
and line numbers referring to the revised manuscript.

**The manuscript submitted by Aga et al. describes the application of the CryoGrid model to simulate ice segregation and thaw consolidation in permafrost. An important factor in determining the impact of warming in permafrost environments is the ground ice content. Efforts to improve the assessment of ground ice content can therefore improve**
**predictions of the impacts of climate change. While the model described in the MS makes some progress in this respect, it appears to be largely limited to segregated ice formation in the upper part of the permafrost, i.e. the transient layer. My main concern with the MS appears to be similar to the point raised by Reviewer 2 that the authors consider a relatively short period and formation of 2-3 cm of segregated ice and do not consider the greater accumulations of ice that formed over longer time periods which would make sense given this is a modelling paper focussing on ice segregation**
**and thaw consolidation. Although the authors refer to scenarios with a spin-up of 1000 years it isn't clear to me they have adequately addressed the concern given the spin up refers to 10-year period repeated several times rather than consideration of the historical climate. My main concern with the MS is the authors do not seem to consider ice segregation that occurred as permafrost formed. In areas that were glaciated in North America for example, permafrost largely formed in the glacial sediments after the glaciers receded so relatively young permafrost exists (the age depending on**
**the effect of subsequent warm and cold periods during the Holocene). Ice segregation would have occurred at the base of the permafrost as freezing progressed, until the ice accumulation was unable to overcome the effective stress. This results in large ice accumulations at depth in the fine-grained glacial and glacial lacustrine sediments for example (to depths of 5-10 m or deeper and not necessarily related to ongoing sedimentation) – see for example, Gaanderse et al. (2018); Wolfe and Morse (2017); Smith et al. (2007). From what I can tell the model does not consider this accumula-**
**tion of ice which is important to consider in assessments of the impact of warming. This would appear to be a rather important limitation to this model.**

We agree with the reviewer that the ground ice distribution in epigenetic permafrost is not only controlled by the conditions of the last 100 to 1000 of years, but by the evolution of the conditions since the permafrost formed. However, we focus on
syngenetic permafrost in this study, assuming frozen conditions at the beginning of the simulation and taking into account sedimentation. This allows us to simulate the ground ice distribution in the uppermost soil layers.

Our model would be in theory also capable of simulating epigenetic permafrost, assuming unfrozen conditions at the beginning. All necessary processes are implemented in the model code to simulate not only ice segregation at the top of the permafrost but also at the base of the advancing freeze front, as suggested by the reviewer. However, this functionality is currently restricted by several factors as discussed in Sect. 4.4: Preparing bias-free forcing data over such long time periods is challenging and beyond the scope of this work. The sedimentation regime including sedimentation rate and type of deposited material would be necessary in case sedimentation occurs at the field site. Furthermore, 3D effects need to be considered such as lateral water fluxes and potential non-horizontal freeze fronts (as occurring during palsa formation). These points limit the possibility to apply the model to epigenetic permafrost, even though the model could perform such simulations. This would be an interesting aspect for future studies.

We included clarifications in the discussion of our manuscript and added a paragraph in the section about limitations.

Line 681 ff.: *Nevertheless, the comparison to observed ground ice contents suggests that our model (with the simplified stratigraphy) is capable of modelling ice segregation in the correct order of magnitude at both the Bayelva field site and Samoylov island. However, we see three main limitations of our model setup which need to be addressed to simulate more realistic cryostratigraphies: (i) the climate data used for spin-up, (ii) the constant sedimentation regime and (iii) the hydrological regime. Overcoming these challenges would allow us to simulate the ground ice evolution since the formation of permafrost, including ice segregation at the permafrost base. This would enable to resolve the ground ice distribution also in greater depths in the soil column, e.g. the ice accumulations in glacial and lacustrine sediments (Gaanderse et al., 2018; Smith et al., 2007; Wolfe and Morse, 2017).*

Line 747 ff.: *In this study, we run the model for syngenetic permafrost, i.e. assuming frozen conditions at the beginning of the simulation and simulating the evolution under a sedimentation regime. Like this, the ground ice distribution in the uppermost parts can be obtained. The ground ice in deeper soil layer could be modelled in principle with an extended spin-up period. In contrast to that, epigenetic permafrost could be simulated through a model setup with unfrozen conditions at the beginning. In this case, ground ice would likely be formed at the top of the permafrost and at the base of the advancing freeze front, but three-dimensional effects such as lateral water fluxes and non-horizontal freeze fronts might play a major role and are not accounted for in the current model. Furthermore, both for syngenetic and epigenetic permafrost, plaeo-climate data to force the model (and potentially information on the sedimentation regime) are required to obtain realistic ground ice distributions (Sect. 4.4).*

**Several conclusions presented were not unknown including the influence of various factors on formation of segregation ice. It has been well known for decades that soil type is important and that ice accumulation is greater in fine-grained material and peat (e.g. Konrad and Morgenstern 1983; Williams and Smith 1989). This is based on field evidence and lab experiments. There was quite a bit of research done on ice segregation and frost heave 30-50 years ago including model development so a great deal of literature exists including that generated by engineers but there appears to be limited consideration of this body of work. This includes the large body of work by RD Miller, as well as Konrad**

**and Morgenstern (1980,1981, 1982 a,b, 1983), O'Neill (1983); Nixon (1991) and others mentioned in the comments be-**
**low (see also ref list).**

We agree with the reviewer that influencing factors on ice segregation and thaw consolidation are known from earlier work. This model includes these processes in the framework of a land surface model, allowing a simulation in dependency on changing climatic conditions. To avoid misunderstandings, we changed the wording in the introduction (research objectives) and the
conclusions as suggested by the reviewer. Please refer to comments below.

We see the point that our manuscript can benefit from more references on earlier work. We included the suggested references in the manuscript. Please refer to the comments below.

**L5 – revision suggested: "...capable of simulating segregated ice formation..."**

We changed the wording as suggested.

Line 4 ff.: *In this study, we present a model scheme, capable of simulating segregated ice formation during a model spin-up*
*together with associated ground heave.*

**L29-30 Note O'Neill et al. (2019) only considered 3 ice types in their model but there are others including injection ice. Reference could be made to the IPA glossary, French (2017) or French and Shur (2010).**

We included injection ice in the manuscript.

Line 29 ff.: *Ground ice can be present as pore ice or excess ice, which can occur as relict ice, wedge ice, segregated ice or injection ice (French and Schur, 2010; French, 2017).*

**L45-46 – There is also field evidence of segregated ice in this type of material, such as information collected from geotechnical boreholes, e.g. , Gaanderse et al. (2018); Wolfe and Morse (2017); Smith et al. (2007).**

We added the information and included the references.

Line 45 ff.: *O'Neill et al. (2019) modelled the occurrence of segregated ice in Canada, showing abundance in fine-grained lacustrine sediments, raised peat plateaus and uplifted marine sediments. This distribution is supported by borehole information and field studies (Gaanderse et al., 2018; Smith et al., 2007; Wolfe and Morse, 2017).*

**L52-53 – This is confusing as the formation of ice releases latent heat which would delay freezing. The effect of latent heat release reduces cooling of the active layer in fall/winter (e.g. Riseborough and Smith (1998).**

Ground ice formation releases latent heat, delaying the freezing. In contrast, the thaw of ground ice consumes energy, delaying the permafrost thaw. This is supported by the study of Riseborough (1990). We clarified this in the manuscript.

Line 51 ff.: *The ground ice content and its distribution strongly determines the sensitivity of permafrost to thaw (Jorgenson et al., 2010; Nitzbon et al., 2019). Ground ice formation releases latent heat, delaying the freezing. In contrast, ice-rich layers in the soil can delay permafrost degradation as energy is consumed upon melting of the ground ice, which is consequently not available for the warming of the ground (Riseborough, 1990).*

**Figure 1 – There is also upward migration of water towards the freezing front at the base of permafrost as it aggrades.**

We changed the figure accordingly.

[Figure]

**Figure 1.** Illustration of ground heave through ice segregation at the top and the base of the permafrost. (b) If the segregated ice is preserved over a long time period, layers with segregated (excess) ice are forming, causing heave of the ground surface. Figure modified after Fu et al. (2022).

**L69-72 – As mentioned above there has been much earlier work done with respect to modelling frost heave (e.g. papers by Konrad and Morgenstern; Nixon 1991 etc.)**

We included the suggested references in the manuscript.

Line 70 ff.: *Thaw consolidation has been the focus of model development for many decades (Morgenstern and Nixon, 1971; Nixon and Morgenstern, 1973; Sykes et al., 1974; Konrad and Morgenstern, 1980, 1981, 1982a, b; Konrad, 1983; O'Neill, 1983; Nixon, 1991; Foriero and Ladanyi, 1995; Dumais and Konrad, 2018)...*

**L75-78 – See previous comment regarding issue of latent heat release.**

We changed the introduction as described in the comment above. Here, we formulated the sentence in a way that could be misunderstood. We changed the formulation.

Line 78 ff.: *Furthermore, as ice segregation is not implemented, they neglect the delay in permafrost warming through the thaw of segregated ice layers, formed during the simulation period.*

**L78 – Revision suggested: "...ground can be simulated with..."**

We changed the wording as suggested.

Line 80 ff.: *In this study, we demonstrate that segregated ice in the ground can be simulated with a climate-dependent spin-up procedure, which aims at reproducing the evolution of ground ice stocks.*

**L88 – See early comment regarding the fact that the role of these factors was not unknown. Isn't it more correct to say that you evaluate the ability of the model to adequately represent these relationships.**

We changed the wording in the manuscript.

Line 89 ff.: *We evaluate the performance of our model to reproduce known controlling factors on ice segregation and thaw consolidation. Particularly, we analyze different climatic conditions (by applying different forcing data sets), the soil type (by using different grain sizes and compositions) and external loads.*

**L117 – There was much earlier work regarding freezing characteristic curves. See examples in Williams and Smith (1989) and also Horiguchi and Miller (1983) and others.**

We agree with the reviewer that there was earlier work on freezing characteristic curves. However, this is the methods section and we describe here on which work our model is based. To include earlier work would confuse the reader in our opinion. Therefore, we decided to not include this literature in the manuscript.

**237-238. There is earlier literature regarding hydraulic conductivity in freezing soils, see examples and figures in Williams and Smith (1989), Horiguchi and Miller (1983), Burt and Williams (1976), Perfect and Williams (1980).**

We added references to the manuscript.

Line 240 ff.: *When the soil freezes, water fluxes are significantly smaller than in unfrozen conditions due to reduced liquid water contents and hydraulic conductivity (Burt and Williams, 1976; Horiguchi, 1983). However, the remaining soil water is still partly mobile. This is mainly driven by matric potentials, which reach considerably negative values for ground temperatures below zero degrees, resulting in an attraction of soil water towards the freezing front (Perfect and Williams, 1980; Williams and Smith, 1989).*

**L248 – Essentially you are only considering one type of excess ice, i.e. segregated ice.**

We agree with the reviewer, that segregated ice is a type of excess ice. At this point in the manuscript, we want to emphasize that also the definition within the model framework is differently. We clarified this in the text.

Line 252 ff.: *We highlight that the term "segregated ice" is defined differently within the framework of the CryoGrid community model than the term "excess ice" in previous versions of the model.*

**L255-256 – Formation of other types of ice are associated with different process eg. Thermal contraction cracking required for ice wedge formation.**

We added this information in the manuscript.

Line 259 ff.: *We note that the new model scheme can only represent segregated ice and that the formation of other forms of excess ice such as wedge ice cannot be accounted for as they are associated with different processes during formation.*

**L272 – As mentioned in general comments, these examples aren't necessarily representative of conditions everywhere with respect to climate and geological history.**

This is a proof-of-concept study to evaluate the performance of the newly developed model scheme. We discuss the challenges regarding the historical climate data in Sect. 4.4 and extended it according to the first comment of the reviewer. Further-
more, we added a paragraph in the section about limitations. Please see the answer to the first comment of the reviewer.

**L326-327 – The model seems to assume that frozen conditions at depth already exist but there is no simulation of the formation of segregated ice as the permafrost initially formed.**

It is currently not possible to simulate the conditions during permafrost formation. This is not because of the presented model scheme but due to the lacking historical forcing data. To prepare such forcing data is a big challenge and out of the scope of this study. We extended parts of the discussion and the section about limitations. Please see the answer to the first comment of the reviewer.

**L481 – revise to "thicker active layer" or "deeper permafrost table"**

We changed the wording.

Line 484 ff.: *Despite less ice segregation, the ground heave is more pronounced in Bayelva during the spin-up, due to wetter*
*conditions and a deeper permafrost table, resulting in stronger soil swelling.*

**L485 – The water migration is dependent on the temperature gradient and thermal conditions will also affect the hydraulic conductivity (see refs provided earlier).**

We added this information in the manuscript and added the suggested references.

Line 488 ff.: *Varying the climatic forcing shows, that the model results depend on both soil moisture and temperature gradients in the ground, controlling the water migration towards the freezing front (Burt and Williams, 1976; Perfect and Williams, 1980).*

**L486-505 – As mentioned in general comments, the role of material type in ice segregation was not unknown and is a key consideration in determinations of frost susceptibility or segregation potential (see papers by Konrad and Morgenstern). There is also much field evidence of occurrence segregated ice in fine-grained soils (see refs in general comments) and permafrost maps showing ground ice content including the circumpolar IPA map or O'Neill et al. (2019) base the**

**ice content on material type.**

We agree with the reviewer that this was known from earlier studies. Here, we aim to represent these processes in our model. Therefore, we changed the wording in our research objectives.

Line 89 ff.: *We evaluate the performance of our model to reproduce known controlling factors on ice segregation and thaw consolidation. Particularly, we analyze different climatic conditions (by applying different forcing data sets), the soil type (by using different grain sizes and compositions) and external loads.*

**L506-517 – Others have considered role of loading on segregation process e.g. Konrad and Morgenstern (1983).**

We added the suggested reference in the manuscript.

Line 512 ff.: *Applying an external load influences the formation and thaw of segregated ice (Konrad, 1983, Fig. 12).*

**L561 – As mentioned in earlier comments the consideration of fixed amount of excess ice at the beginning of the simulation is an important limitation of the model. There is no consideration of formation of segregated ice as the freezing front progresses into the soil as permafrost forms.**

While the model is capable of simulating the conditions during permafrost formation, including ice segregation as the freez-
ing front progresses into the soil, we cannot do any model simulations for this scenario yet due to lacking historical forcing data. We extended parts of the discussion and the section about limitations. Please see the answer on the first reviewer comment.

**L693 – There were many of these experiments at bench and field scale in the past and reported in engineering litera-ture (Konrad and Morgenstern papers cited above may include some) and there is also work done by the Geotechnical**
**Science Laboratory at Carleton University in the 1980s and 1990s both bench scale and field scale at facility in Caen (some eg. Smith and Onysko; Williams and Wood 1985; Compendium of reports related to Caen facility, i.e. Canada-France ground freezing expt. can be found in Smith and Burgess 2007).**

We included more references in the manuscript.

Line 701 ff.: *Another possibility for validation could be laboratory freezing experiments. An example is the study by Xue et al. (2021), who conducted a one-sided freezing experiment in saturated soil to investigate the relationship of matric potential, unfrozen water content and segregated ice. Further experiments have been presented by Konrad and Morgenstern (1980, 1981,*

*1982a, b);Smith and Onysko (1990); Williams and Wood (1985).*

**L700 – See earlier comment regarding lack of consideration of ice accumulation at permafrost base as permafrost forms.**

Such simulations would require bias-free historical forcing data and information about the sedimentation regime. We ex-
tended parts of the discussion and the section about limitations. Please see the answer to the first comment of the reviewer.

**L724 – There was earlier literature on role of creep in segregation processes (might be in some pubs I've already mentioned or in body of work by RD Miller).**

We included the reference of Williams and Smith (1989), which was suggested by the reviewer, in the manuscript.

Line 730 ff.: *Creep processes can play an important role in permafrost and can occur at very low slope angles (Williams and Smith, 1989). The soil mechanical processes implemented in the model consider only primary consolidation, and do not account for long-term creep processes, which can be considered to be a first order approximation. For long-term simulations*
*with thick sedimentary deposits near or above thawing temperatures, creep processes should be implemented to get a better representation of the deformation of the soil column.*

**L744-747 – Considering this period or other earlier cooling periods would require consideration of formation of permafrost at base of permafrost as the frost front progresses.**

This process can be simulated with the model. However, we did not perform simulations including permafrost formation due to lacking paleo-forcing data and information about the sedimentation regime. We extended parts of the discussion and the section about limitations. Please see the answer to the first comment of the reviewer.

**L772-778 – See previous comments regarding the fact that most of this was not unknown so we didn't require the model to suggest them. It is more correct to say that you assessed the ability of the model to consider the relationship of these factors to ice segregation.**

We agree with the reviewer and changed the manuscript accordingly.

Line 790 ff.: *The model shows that important controlling factors on ice segregation and thaw consolidation can be simulated such as (i) ground temperature gradients and soil water content, (ii) soil type and (iii) external loads.*

**References**

**Burt T and Williams PJ 1976. Hydraulic conductivity of frozen soils. Earth Surface Processes 1 (3): 349-360.** Included in the manuscript.

**French HM 2017. The periglacial environment. 4th Edition.** Included in the manuscript.

**Gaanderse AJR et al. 2018. Composition and origin of a lithalsa related to lake-level recession and Holocene terrestrial emergence, Northwest Territories, Canada. Earth Surface Processes and Landforms, Composition and origin of a lithalsa related to lake-level recession and Holocene terrestrial emergence, Northwest Territories, Canada, 43, 1032–1043 DOI: 10.1002/esp.4302.** Included in the manuscript.

**Horiguchi, K and Miller, RD 1983. Hydraulic conductivity functions of frozen materials., 504-508.** Included in the manuscript.

**Konrad, JM and Morgenstern, NR 1983. Frost susceptibility of soils in terms of their segregation. Proc. 4th Int. Conf. on Permafrost, Fairbanks AK, 660-665.** Included in the manuscript.

**Konrad, JM and Morgenstern, NR 1980. A mechanistic theory of ice lens formation in fine-grained soils. Canadian Geotech. J. 17:473-486.** Included in the manuscript.

**Konrad, JM and Morgenstern, NR 1981. The segregation potential of a freezing soil, Can. Geotech. J. 18:482-491.** Included in the manuscript.

**Konrad, JM and Morgenstern, NR 1982a. Prediction of frost heave in the laboratory during transient freezing, Can. Geotech. J. 19:250-259.** Included in the manuscript.

**Konrad, JM and Morgenstern, NR 1982b. Effects of applied pressure on freezing soils. Can. Geotech. J. 19: 494-505.** Included in the manuscript.

**Nixon, JF 1991. Discrete ice lens theory for frost heave in soils. Can. Geotech. J. 28:843-859.** Included in the manuscript.

**O'Neill, K. 1983. The physics of mathematical frost heave models: a review. Cold Reg. Sci and Tech 6:275-291.** Included in the manuscript.

**Perfect, E and Williams PJ 1980. Thermally induced water migration in frozen soils. Cold Reg. Sci and Tech. 3: 101-109.** Included in the manuscript.

**Riseborough, D.W., and Smith, M.W. 1998. Exploring the limits of permafrost. In Proceedings of Seventh International Conference on Permafrost. Yellowknife, Canada. June 1998. Collection Nordicana Vol.57, pp. 935-941.** We used instead: Riseborough, D.W. (1990): Soil latent heat as a filter of the climate signal in permafrost.

**Smith MW and Onysko D 1990. Observations and significance of internal pressures in freezing soil. Proc. 5th Canadian Permafrost Conf. Collection Nordicana No. 54, p. 75-81. http://pubs.aina.ucalgary.ca/cpc/CPC5-75.pdf.** Included in the manuscript.

**Smith, S.L., Ye, S., and Ednie, M. 2007. Enhancement of permafrost monitoring network and collection of baseline environmental data between Fort Good Hope and Norman Wells, Northwest Territories. Geological Survey of Canada Current Research, 2007-B7: 10. doi:10.4095/224524.** Included in the manuscript.

**Smith, S.L., and Burgess, M.M. (compilers) 2007. Compendium of Reports and Databases Produced Under the Canada-France Ground Freezing Experiments. Geological Survey of Canada Open File 5593. https://doi.org/10.4095/223900.** Not included in the manuscript as other references fitted better.

**Williams PJ and Smith MW 1989. The Frozen Earth: fundamentals of geocryology. Cambridge, 306p.** Included in the manuscript.

**Williams, PJ and Wood JA. 1985. Internal stresses in frozen ground. Canadian Geotechnical Journal 22: 413-416 https://doi.org/10.1139/t85-054.** Included in the manuscript.

**Wolfe, SA, Morse PD 2017. Lithalsa Formation and Holocene Lake-Level Recession, Great Slave Lowland, Northwest Territories. Permafrost and Periglacial Processes, 28: 573–579 DOI: 10.1002/ppp.** Included in the manuscript.